# How many modes are needed to predict climate bifurcations? : Lessons from an experiment

Bérengère Dubrulle [1], François Daviaud [1], Davide Faranda[2,3,4], Louis Marié[5], and Brice Saint-Michel[6]

[1]Université Paris-Saclay, CEA, CNRS, SPEC, CEA Saclay 91191 Gif sur Yvette cedex, France
[2]Laboratoire des Sciences du Climat et de l'Environnement, UMR 8212 CEA-CNRS-UVSQ, Université Paris-Saclay, IPSL, 91191 Gif-sur-Yvette cedex, France
[3]London Mathematical Laboratory, 8 Margravine Gardens London, W6 8RH, UK
[4]LMD/IPSL, Ecole Normale Superieure, PSL research University, 75005, Paris, France.
[5]LOPS, UMR6523, Univ. Brest, CNRS, IFREMER, IRD, 29280, Plouzané, France
[6]Department of Chemical Engineering, Delft University of Technology, Van der Maasweg 9, 2629HZ, Delft, the Netherlands

**Correspondence:** B. Dubrulle (berengere.dubrulle@cea.fr)

**Abstract.** According to everyone's experience, predicting the weather reliably over more than 8 days seems an impossible task for our best weather agencies. At the same time, politicians and citizens are asking scientists for climate projections several decades into the future to guide economic and environmental policies, especially regarding the maximum admissible emissions of $CO_2$. To what extent is this request scientifically admissible?

In this review we will investigate this question, focusing on the topic of predictions of transitions between metastable states of the atmospheric or oceanic circulations. Two relevant exemples are the switching between zonal and blocked atmospheric circulation at midlatitudes and the alternance of El Niño and La Niña phases in the Pacific ocean. The main issue is whether present climate models, that necessarily have a finite resolution and a smaller number of degrees of freedom than the actual terrestrial system, are able to reproduce such spontaneous or forced transitions. To do so, we will draw an analogy between climate observations and results obtained in our group on a laboratory-scale, turbulent, von Kármán flow, in which spontaneous transitions between different states of the circulation take place. We will detail the analogy, and investigate the nature of the transitions, the number of degrees of freedom that characterizes the latter and discuss the effect of reducing the number of degrees of freedom in such systems. We will also discuss the role of fluctuations and their origin, and stress the importance of describing very small scales to capture fluctuations of correct intensity and scale.

## 1 Context

The present review paper is based on the lecture delivered by Bérengère Dubrulle, at the occasion of her reception of the Lewis Fry Richardson Medal 2021. The story around this lecture started back in the year 2000, when Bérengère became interested in climate change and started discussing with colleagues at *Laboratoire des Sciences du Climat et de l'Environnement*, (LSCE), she was intrigued by a strange behavior of the temperature curves discussed in the IPCC reports: they all exhibited a constant, quasi-linear increase with time, linearly following the rise of $CO_2$ concentration. Given that the perturbation of the $CO_2$ concentration was far from being negligible (we were talking about doubling it), and that climate is a highly nonlinear system, she

was wondering why the output of climate models did not show at the time any sign of a non-linear response. It was surprising as her research group routinely observed non-linear behavior such as bifurcations or saturations in experimental set-ups or in numerical simulations of turbulent flows.

She became all the more puzzled as the climate community was starting to acknowledge the possibility of occurrence of tipping points (Russill and Nyssa, 2009; Russill, 2015). In particular, some of her frequent interlocutors like Didier Paillard, Pascal Yiou or Gilles Ramstein pointed to her interesting occurrences of bifurcations in low dimensional models of the atmosphere (e.g. the Stommel model (Stommel, 1961)) or spontaneous abrupt transitions in (proxies of) temperature records during the last hundreds of millenia (see figure 1). More recently, multi-stability and bifurcations were also observed in a simplified

climate model (Margazoglou et al., 2021).

On the other hand, the climate science community has achieved considerable success in predicting the increase of Earth average temperatures in the last few decades. This success shows that present climate models, though still incomplete and perfectible, capture the correct evolution of atmospheric or oceanic circulation, at least in terms of large scales features and most probable events. Recent versions of climate models also include non-linear effects on oceanic circulation (Boucher et al.,

2020), vegetation (Guimberteau et al., 2018) and ice-sheet dynamics (Charbit et al., 2019). They also better represent the energy and water cycles, including more physical constraints on conservation laws (Irving et al., 2021). They are therefore able to capture nowadays some non-linear interactions between the different components of the climate system, i.e. the El Nino Southern Oscillation (ENSO) feedback to atmospheric motions (Bayr et al., 2020), monsoons (Yang et al., 2019), or stratospheric to tropospheric interactions (Olsen et al., 2007). Geophysicists are therefore studying extensively the non-linear

properties of the different components of the climate system (atmosphere, hydrosphere, litosphere, biosphere and cryosphere) that appear ubiquitously.

In the atmosphere, characterizing the geometry and the dynamics of the polar jet is still an open problem, as it can be in an almost zonally symmetric state with strong zonal currents associated with trains of extratropical cyclones, or in broken, flower-like, so-called "blocked" states which induce heat or cold-waves depending on the season and the geography (Serra

et al., 2017). Despite the great increase of computational power of climate models, the switching between the two main phases of the jet is still difficult to characterize statistically and dynamically (Faranda et al., 2019b). It is therefore complicated to determine its evolution under climate change.

In fact, non-linear phenomena arise in all the "spheres". In the ocean, while ENSO is reproduced by most models, correctly reproducing the magnitude and frequency of its occurrence is still challenging, and the fate of the thermohaline circulation

remains to be determined; whereas the biosphere is a mine of non-linear interactions between living species directly breathing the non-linear atmospheric chemistry and reacting non-linearly (losing leaves, migrating, hibernating, ... ) to changes in their physical environment.

Considering all that, it appeared clear to Bérengère that the richness of non-linear interactions in climate needed to be further understood and she had a feeling that laboratory experiments of turbulence could help guide intuition regarding the role of such

non-linearities in climate bifurcations, as already demonstrated earlier by Weeks et al. (1997) and the work of Hide (Ghil et al., 2010).

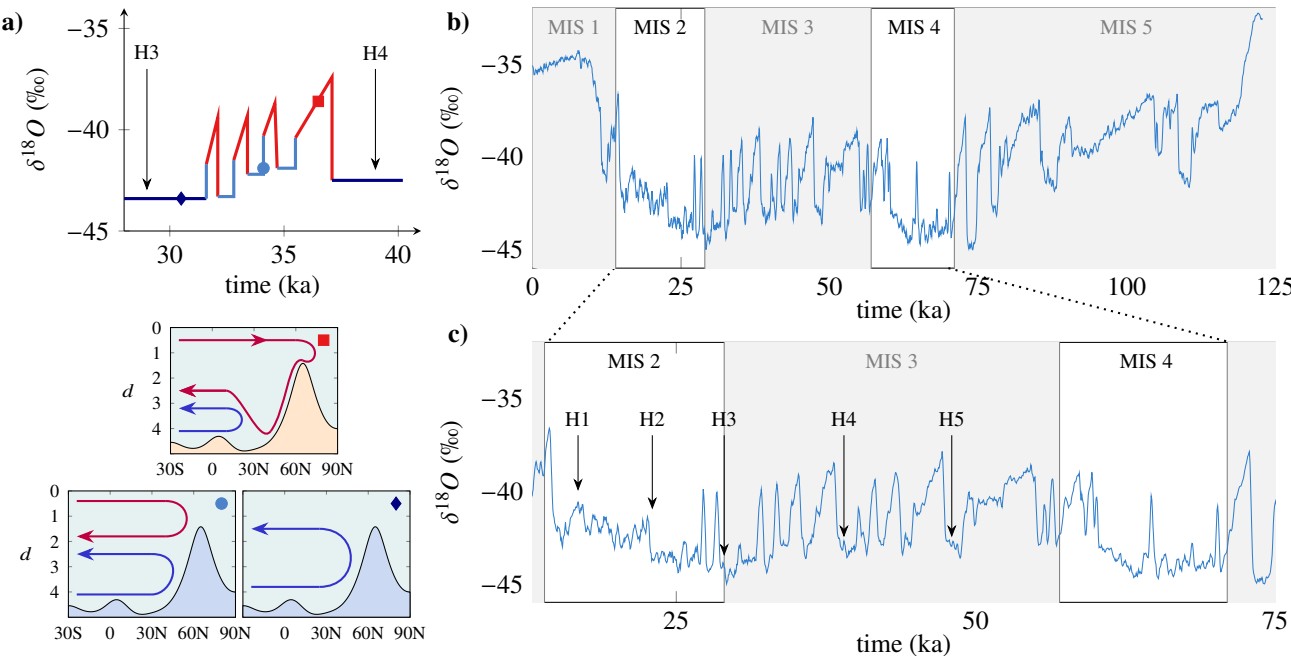

**Figure 1.** (a) Diagram of a Dansgaard-Oeschger event, alternating hot (red) and cold (blue and sky blue) periods, some of which coincide with a massive iceberg debacle (numbered with an H). A reconstruction of the ocean circulation during this period (provided in the insets) shows that each phase of this cycle is associated with a different oceanic circulation (Sarnthein et al., 1994). (b) Recording of the rapid climate variability of the last ice age obtained by the $\delta^{18}O$ isotope profile for the Greenland core. The x-axis represents the time spent (in ka). Our present time is located on the origin of this axis. We indicate the first five Marine Isotope Stages (MIS) in the graph. Traditionally, the temperature is reconstructed from a relationship between temperature and $\delta^{18}O$. However, many biases such as the seasonality of precipitation or the temperature of regions of evaporation of water masses lead to an underestimated reconstructed temperature. Thus the temperature difference between the current period and the last glacial maximum at 21,000 years is estimated at $20°C$ while a Dansgaard-Oeschger (numbered) event is estimated between $10°C$ and $16°C$. Adapted from Andersen (2004).

This review is the story of the long journey she undertook with four main collaborators into theoretical, laboratory and numerical explorations, to try and understand these mysterious apparent contradictions, and examine how many modes it takes to capture transitions in climate models.

## 2  How many degrees of freedom are theoretically and practically involved in climate simulations?

Simulating climate is an arduous task, for we need to describe the interaction of fluid envelopes (atmosphere, ocean) with the lithosphere, cryosphere, biosphere under solar forcing. This makes the climate a non-equilibrium non-linear and complex system. Complex here means that there are several interacting scales at which the energy is accumulated and distributed in space, time and towards other scales through energetic processes that are visible for humans as meteorological and oceanic

phenomena, e.g. cyclones, thunderstorms, marine currents, iceberg break-off. How many degrees of freedom are needed to take into account this complexity is still an open question. The number we choose is therefore fixed by necessity (i.e. by computing capabilities) , rather than by reason. Let us take for example the case of the fluid envelopes.

Their basic physics obey the non-linear, partial differential equations proposed by Navier and Stokes two hundred years ago. They describe the dynamics of a velocity field $u(x,t)$ under the action of a pressure field, viscous dissipation and stirred by volume forces. From the work of Kolmogorov on turbulence (Kolmogorov, 1941), hereafter referred to as the K41 theory of turbulence, we know that the balance between forcing and dissipation results in a self-similar organization of the fluid. Kinetic energy is injected at scale $L_f$ and is transferred at a constant rate $\epsilon$ by the energy flux down to the scale $\eta$ where the energy flux becomes compensated by the viscous flux, with $\eta = (\nu^3/\epsilon)^{1/4}$ and $\nu$ is the viscosity. At smaller scales, the energy flux is transported by the viscous processes to the smallest hydrodynamic scale, where it is dissipated into heat. A turbulent flow then displays vortices of all sizes in between $L_f$ and $\eta$, and its energy spectrum scales like $E(k) \sim \epsilon^{2/3} k^{-5/3}$. This means that if we want to capture the flow physics (e.g; location of vortices and their dynamics in the flow, energy dissipation, interplay between vortices at different scales), we need to discretize the Navier-Stokes equations on a grid that is $(L_f/\eta) \times (L_f/\eta) \times (L_f/\eta)$, containing *a priori* $N \sim (L_f/\eta)^3$ degrees of freedom (as we shall discuss later, the actual number may even be larger!) . If we put numbers corresponding to the atmosphere ($L = 10^3$ km, $\eta = 10$ mm), and take into account the anisotropy (Schertzer and Lovejoy, 1991), we get an astronomical number: $N \sim 10^{27}$, larger than the Avogadro number. Reading-in or writing out this volume of data at each time-step to advance the flow would take 73 billion years of CPU time at the pace of the fastest massively parallel computers!

In recent years there have been developments in understanding that this computational nightmare can be partially solved by applying neural networks or, more generally, machine learning approaches (see, e.g. Pathak et al. (2017, 2018) for artificial intelligence methods applied to the behavior of chaotic systems). We would like to stress that these approaches are never holistic and often target a specific subset of spatial and temporal scales of the climate systems, e.g. the prediction of geophysical data (Wu et al., 2018), the parameterizations of subgrid processes in climate models (Krasnopolsky et al., 2005; Krasnopolsky and Fox-Rabinovitz, 2006; Rasp et al., 2018; Gentine et al., 2018; Brenowitz and Bretherton, 2018, 2019; Yuval and O'Gorman, 2020; Gettelman et al., 2020; Krasnopolsky et al., 2013), the forecasting (Liu et al., 2015; Grover et al., 2015; Haupt et al., 2018; Weyn et al., 2019; Faranda et al., 2021)and nowcasting (i.e. extremely short-term forecasting) of weather variables(Xingjian et al., 2015; Shi et al., 2017; Sprenger et al., 2017), the quantification of the uncertainty of deterministic weather prediction (Scher and Messori, 2018). The greatest challenge of entirely replacing the equations of climate models with neural networks capable of producing reliable long and short-term forecasts of meteorological variables is, to the best of our knowledge, not yet achieved with these methods.

We are then led by necessity to simulate much fewer degrees of freedom, typically a few thousands in the atmosphere or ocean for recent climate models. How reasonable is this drastic reduction of the number of degrees of freedom? It now depends on the flow physics: the self-similar energy spectrum is an indication that some scales or modes may play a more prominent role than others. So, maybe, the theoretical $N \sim (L_f/\eta)^3$ figure overestimates the actual number of modes that is needed to represent the flow dynamics, and we could circumvent the computational obstacle by a clever selection of grid points or modes.

In 1963, a pioneer study by Lorenz (1963) showed that only 3 modes were necessary to obtain chaotic behavior on an attractor and reproduce some key dynamical effects observed in Rayleigh-Bénard convection, and essential ingredient of atmospheric dynamics. More generally, it is thought that if we were able to understand what the "attractor" of the climate dynamics is, then the number of modes we would need is simply the dimension of such attractor. However, both the computation of the attractor for climate, and the identification of the independent variable sufficient to describe it is still a highly debated issue that remains far from being solved theoretically (Faranda et al., 2019a; Falasca et al., 2019; Brunetti et al., 2019).

Climate models therefore use a simpler idea and currently select modes based on length scales: the idea is that the largest scales are very energetic, and do not directly feel the viscosity, which becomes active only at scales of the order of $\eta$. The large scales are however indirectly influenced by the viscosity through their interaction with the small scales, that play the role of a "turbulent viscosity" with respect to the large scales. The concept of turbulent viscosity is therefore akin to a clever renormalization procedure, that takes care of the energy accumulation taking place at the cut-off scale (Forster et al., 1977; Frisch et al., 1980; Herring et al., 1982) and reference therein. Hence, current climate models only keep the largest modes, and add a "turbulent viscosity" to take into account the influence of the small scales. The magnitude of the turbulent viscosity is known to depend on the cut-off scale since Richardson (Richardson and Walker, 1926) and is usually significantly higher than the classical Newtonian viscosity of the fluids under study: for a cut-off scale of the order of a few hundred of kilometers- the smallest scale that climate models can currently solve without saturating memories of modern super-computers-, turbulent viscosities are equivalent to the Newtonian viscosity of tar in the atmosphere, while in the ocean, they are equal to that of honey.

Replacing air and water by tar and honey does not look too appealing. Moreover, there are a lot of more complex processes like beating / backscatter / intermittency (Forster et al., 1977; Frisch et al., 1980; Herring et al., 1982) that cannot be taken into account by a simple "turbulent viscosity". Yet, numerical simulations based on these assumptions provide a fairly realistic picture of the past and present state of circulation in the ocean and in the atmosphere (Flato et al., 2013). There is therefore a rationale in this procedure that we should try to understand and improve. To do so, we considered a laboratory model experiment that retains some essential properties of natural flows (forcing, dissipation, symmetries, wide range of spatial and temporal scales) but with simple boundary conditions, and few interacting components, in order to limit the number of relevant observables to be analyzed. To some extent, this is similar to the simplification obtained by considering a perfect monoatomic gas to describe a real gas.

## 3 The von Kármán flow, an analogue for oceanic or atmospheric large scale circulation

Our experimental setup is summarized in Fig. 2: we consider a transparent cylindrical tank of aspect ratio 1.8, filled with water. The tank is closed at each end by two coaxial and counter-rotating impellers with curved blades. Such a device provides a very efficient, inertial stirring of the liquid inside the tank, which easily reaches speeds of the order of 1 m/s for an impeller rotation frequency of 10 Hz in a 10 cm radius tank. The flow produced in the device (called von Kármán flow) is highly turbulent,

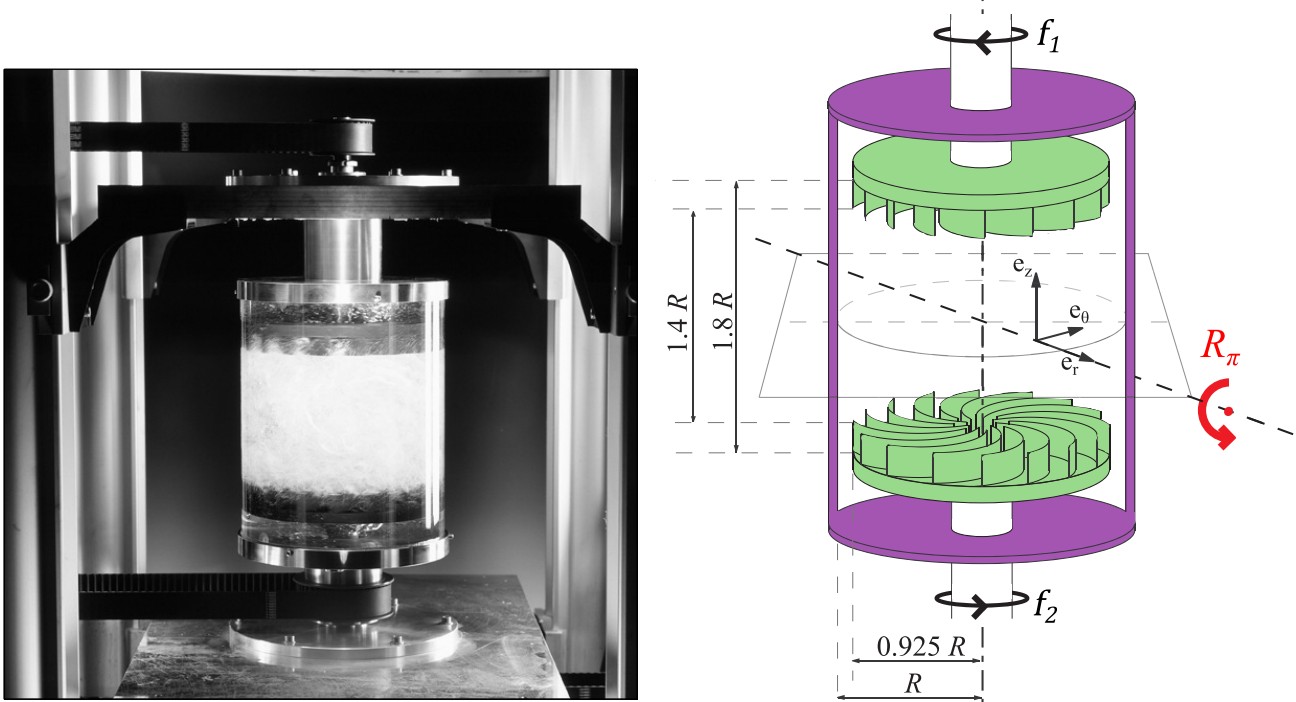

**Figure 2.** Picture and diagram of the experimental set-up. The black arrows indicate the direction of turbine rotation. Symmetry : the system is symmetric by any rotation $\mathcal{R}_\pi$-an angle $\pi$ (shown by the red arrow) around any axis located in the equatorial plane and passing through the axis of rotation-it is the symmetry of an hourglass that is reversed. Positive vertical angular momentum is injected at the bottom, and negative vertical angular momentum is injected at the top via the impellers. The resulting imbalance induces a large scale circulation, from the top and bottom to the middle plane, in analogy with the large scale oceanic or atmospheric circulations from the equator to the poles. [Picture courtesy Pierre Cortet].

characterized by a Reynolds number in water around $10^6$ - about 1000 times lower than in the atmosphere. We characterize the flow using two complementary types of measurements:

1. The torque or velocity applied by each of the impellers onto the turbulent flow. These are global measurements of the state of the system.

2. Local velocity maps obtained using stereoscopic particle image velocimetry (s-PIV). These measurements give us access to the three instantaneous components of the velocity field in a vertical plane crossing the center of the experiment with an acquisition rate of about 15 Hz. This rate is insufficient to resolve in time all turbulent scales but offers reliable statistics on the velocity field, obtained by averaging several thousands of instantaneous maps.

Averaging a large number of instantaneous velocity fields during statistically steady regimes allows us to observe the mean flow established in the geometry. It takes the form of a large-scale circulation carrying the vertical angular momentum from

**Table 1.** Analogy between the Earth system fluid envelopes and the von Kármán turbulent swirling flow stirred by impellers (Figure 2).

| | Natural flows | von Kármán flow | |
|---|---|---|---|
| Geometry | Spherical | Cylindrical | |
| Energy source | Solar radiation | Impeller rotation | |
| Energy sink | Long-wave radiation | Viscous dissipation | |
| Transported quantity | Heat | Vertical angular momentum | |
| Fixed parameter | Solar constant | Mean impeller rotation rate $(f_1 + f_2)/2$ | Mean impeller torque $(c_1 + c_2)/2$ |
| Perturbations | $CO_2$ concentration | Impeller rotation rate imbalance $\theta = (f_1 - f_2)/(f_1 + f_2)$, | Impeller torque imbalance $\delta = (c_1 - c_2)/(c_1 + c_2)$, |
| | | Impeller geometry | |
| Observables | Global mean $T$ | Mean and imbalance angular momentum flux $(c_1 + c_2)/2$, $\delta = (c_1 - c_2)/(c_1 + c_2)$, | Mean and imbalance impeller rotation rate $(f_1 + f_2)/2$, $\theta = (f_1 - f_2)/(f_1 + f_2)$, |
| | Equator-Pole $\Delta T$ | Global Vertical Angular momentum $I = \int r u_\theta \, \mathrm{d}V$ | |
| Fluids | Air, Water | Water | |
| Reduced system equivalent | Tar, Honey | Glycerol | |
| *A priori* degrees of freedom | $N \sim 10^{24}$ | $N \sim 10^{13}$ | |
| Reduced system degrees of freedom | $N_r \sim 10^4$ | $N_r \sim 10^4$ | |

one impeller to the other. This circulation is analogous to the atmospheric or thermohaline circulation, transporting heat from the equator to the pole, under the action of either solar radiative forcing, and/or surface forcing by winds. The analogy is summarized in Table 1.

We have used our von Kármán setup in two complementary forcing modes. In the first mode, the rotation rates of the impellers are kept fixed. In this case one can set the mean rotation rate of the impellers, $(f_1 + f_2)/2$, but also the normalized rotation rate difference of the impellers, $\theta = (f_1 - f_2)/(f_1 + f_2)$. The former quantity governs the total angular momentum- along the vertical axis- present in the fluid. The latter parameter offers a fine control of the degree of $\mathcal{R}_\pi$-symmetry of the flow forcing. Such a forcing mechanism is analogous to convective Rayleigh Benard systems with fixed temperature gradient.

However, in natural systems, the forcing is rather done by imposing the heat flux that traverses the system (due to solar radiation). We have thus also performed experiments in a second forcing mode, in which we controlled the torque applied to the von Kármán flow by the impellers. In this mode, the mean torque $(c_1 + c_2)/2$ is a measure of the vertical angular momentum flux from top to bottom, while the reduced difference $\gamma = (c_1 - c_2)/(c_1 + c_2)$, which prescribes the dissipation of angular momentum at the cylinder wall, also controls the level of $\mathcal{R}_\pi$-symmetry of the forcing mechanism. In principle, a

mixed forcing type in which the rotation rate of one impeller and the torque of the other one are fixed is also possible, but we have not yet explored the behavior of the system in this case.

One can control the degree of perturbation in the von Kármán flow by controlling the level of velocity fluctuations, for a given set of forcing conditions and viscosity. This may be achieved by changing the curvature of the impeller blades, which will be used as an analogue of changes in $CO_2$ concentration in natural flows.

We can also change the number of degrees of freedom in the von Kármán flow by considering more or less viscous fluids, going from $N \sim 10^{13}$ for a water filled experiment at $Re \sim 10^6$ down to $N \sim 10^4$ for a glycerol filled experiment at $Re \sim 10^2$ .This is an experimental equivalent of the reduction of the number of degrees of freedom that is explicitly done when using turbulent viscosities in present numerical simulations of atmospheric or oceanic circulation. Eventually, the only major difference between natural flows and the von Kármán flow is the symmetry of the geometry, which is spherical rather than cylindrical. From the point of view of a theoretical physicist, though, this difference only affects the detailed shape of the circulation.

## 4  Large scale circulation properties

### 4.1  Flow symmetries and circulation topology

In the von Kármán system, vertical angular momentum of opposite signs is injected by the top and at the bottom impellers. The resulting imbalance produces a large scale meridional circulation analogous to the large scale oceanic and atmospheric circulations between the Equator and the poles. The topology of the large-scale circulation is strongly influenced by the symmetries of the experimental set-up. Arbitrary rotations around the cylinder axis obviously leave the experimental set-up invariant. Its symmetry group thus contains the symmetry group of the oriented circle, the special orthogonal group $SO(2)$. This symmetry usually carries over, in the time-averaged sense, to the large scales of the flow. Another basic symmetry of the system is the $\mathcal{R}_\pi$ symmetry, which exchanges the two impellers by rotation of 180 degrees around any axis of the middle plane which intersects the cylinder axis (the "hourglass reversal" symmetry, see Fig. 2). When the two impellers rotate at exactly the same frequency $f_1 = f_2$, corresponding to a relative speed difference $\theta = (f_1 - f_2)/(f_1 + f_2) = 0$, the system is strictly symmetric with respect to $\mathcal{R}_\pi$ and its symmetry group is the (general) orthogonal group $O(2)$. When $\theta \neq 0$, $\mathcal{R}_\pi$ is broken, the experiment symmetry group reduces to the special orthogonal group $SO(2)$. The latter is the (connected) component of $O(2)$ whose elements have +1 determinants, the other component having determinants -1. This explains how $O(2)$ reduces to $SO(2)$ with the breakdown of rotational symmetry. Except on "Aqua-planets"(Ferreira et al., 2011), the symmetry group of the continents is trivial. If one neglects the continental influences, the symmetry group of the atmosphere is $SO(2) \times \mathbb{Z}_2$ (rotations around the polar axis and mirror-symmetry with respect to the Equator), while the symmetry group of each hemisphere taken separately is $SO(2)$. These symmetry considerations guide us in our search for analogies between the von Kármán system and the Earth system: in some respects, the symmetry group of the $\mathcal{R}_\pi$-asymmetric von Kármán flow is the same as the symmetry group of the atmosphere in a single hemisphere, and insight gained in the study of vertical angular momentum transport in the von Kármán flow may be expected to carry over to the transport of heat from the Equator to one pole in the natural atmosphere.

However, it is well known that the distinct distribution of continental masses between the two hemispheres, or the insulation distribution on seasonal time scales, break the mirror-symmetry of the Earth with respect to the Equator. The $SO(2) \times \mathbb{Z}_2$ symmetry of the Earth system is thus in fact quite imperfect, and in some respects closer to $SO(2)$ symmetry of the von Kármán flow in $\mathcal{R}_\pi$-asymmetric conditions. It can thus be hoped that some aspects of the von Kármán flow could also bear resemblance to phenomena involving the Earth system as a whole, such as the seasonal motion of the Intertropical Convergence Zone at the boreal summer/austral summer transitions, the oceanic thermohaline circulation or the glacial/interglacial climate transitions.

The different large-scale flow configurations observed in the von Kármán flow in the rotation rate-controlled mode when $\theta$ is varied are represented in Figure 3. For large negative (Fig. 3b) or positive (Fig. 3c) values of $\theta$, $\mathcal{R}_\pi$ is unambiguously broken, and the flow is in a $SO(2)$ configuration consisting of a single cell rotating in the direction of the fastest impeller (the analogue of a direct Equator-pole atmospheric meridional circulation in a single hemisphere, of an austral or boreal summer with grossly exaggerated inter-hemisphere asymmetry, or of a glacial episode). In this case the total mean vertical angular momentum $\langle I \rangle = \langle \int r u_\theta \, \mathrm{d}V \rangle$ follows the sign of $\theta$. Negative $\langle I \rangle$ correspond to $b_-$ states ("austral summer", henceforth "A" states), whereas positive $\langle I \rangle$ correspond to $b_+$ states ("boreal summer", henceforth "B" states). The B state obtained for a value of $\theta$ is transformed by $\mathcal{R}_\pi$ into the A state obtained for $-\theta$.

When $\theta$ becomes close to 0, the configuration obtained is closer to $\mathcal{R}_\pi$-symmetry: it consists of two toroidal recirculation cells arranged on either side of the median plane, rotating in opposite azimuthal directions (Fig. 3a). The two halves of the von Kármán flow are almost $\mathcal{R}_\pi$-symmetric images of one another and the total mean vertical angular momentum is close to zero: $\langle I \rangle \simeq 0$. In a way, the situation can also be considered to resemble the global Earth system, in which the two hemispheres are approximately image of one another by reflection across the Equatorial plane, separated by an interface that shifts gradually as the symmetry of the forcing is varied (in analogy to the seasonal cycle of insulation, which brings the latitude of maximal solar energy input into the Earth system alternatively in the two hemispheres). This situation is thus reminiscent of a "spring" or "autumn" transition situation (henceforth "T" state), or of an interglacial situation.

But the situation in that case can also be considered to be analogue to the atmospheric situation in a single hemisphere: it consists of two toroidal recirculation cells arranged on either side of a transition plane, separated by a highly turbulent shear layer - a configuration somewhat equivalent to the Hadley circulation of the atmosphere: positive vertical angular momentum fluid is confined to the lower half of the cylinder, negative vertical angular momentum fluid is confined to the upper half, and vertical angular momentum transport between the two is only mediated by the turbulent fluctuations, which have a visual appearance strikingly similar to that of synoptic-scale atmospheric disturbances. This situation prevails for small values of the impeller rotation rate imbalance $\theta$, for which the mid-plane shear layer (analogue of the midlatitudes "storm track") shifts continuously in "latitude" as $\theta$ varies, moving gradually away from the fastest-rotating impeller.

## 4.2 Seasonal cycle in water experiment

Given the properties of the large-scale circulation, we are therefore able to drive the equivalent of a "seasonal cycle" in our experiment by modulating the forcing asymmetry as a function of time $\theta(t)$. This modulation reflects the yearly modulation of

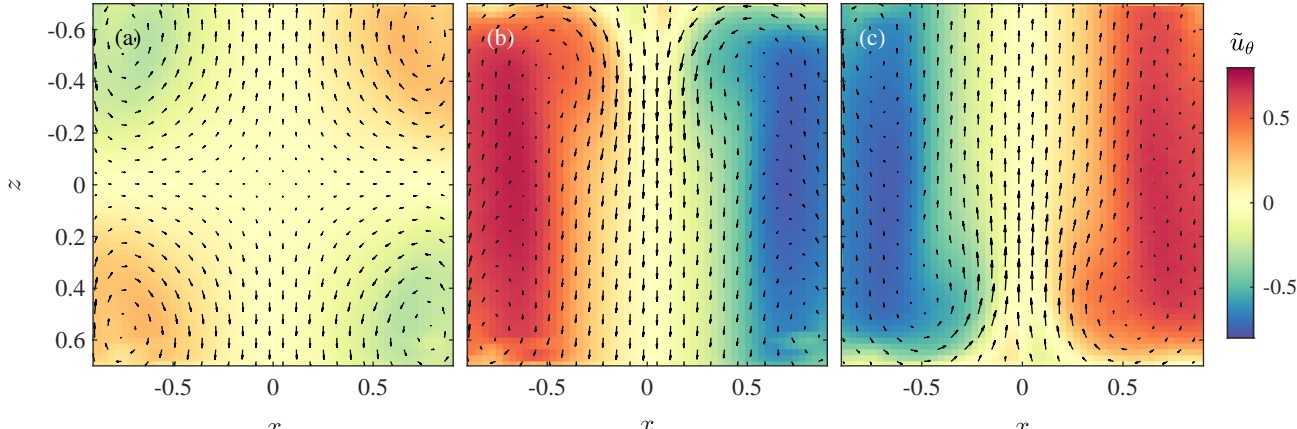

**Figure 3.** Different states observed in our system: (a) Two-cell symmetric state ("spring" or "autumn" transition, or "T", state ) (b) One-cell asymmetric state $b_+$ ( "austral summer", or "A", state). (c) One-cell asymmetric state $b_-$ ( "boreal summer", or "B", state). The normalized azimuthal flow $\tilde{u}_\theta = 2u_\theta/R(f_1 + f_2)$ is shown in color (from negative blue to positive red), while the radial and vertical speeds $(u_r, u_z)$ are represented by arrows. The resolution of the fields has been degraded for better visibility. The positions $x$ and $z$ have been rescaled by the radius of the cylinder $R$. The higher amplitudes in case (b) and (c) with respect to case (a) reflects a fundamental energetic difference, linked to the very strong shear layer appearing at the interface between the flow, rotating in the direction of the dominant impeller, and the slave impeller, that rotates otherwise. As a result, the energy dissipation is 4 times larger in case (b) and (c) with respect to case (a).

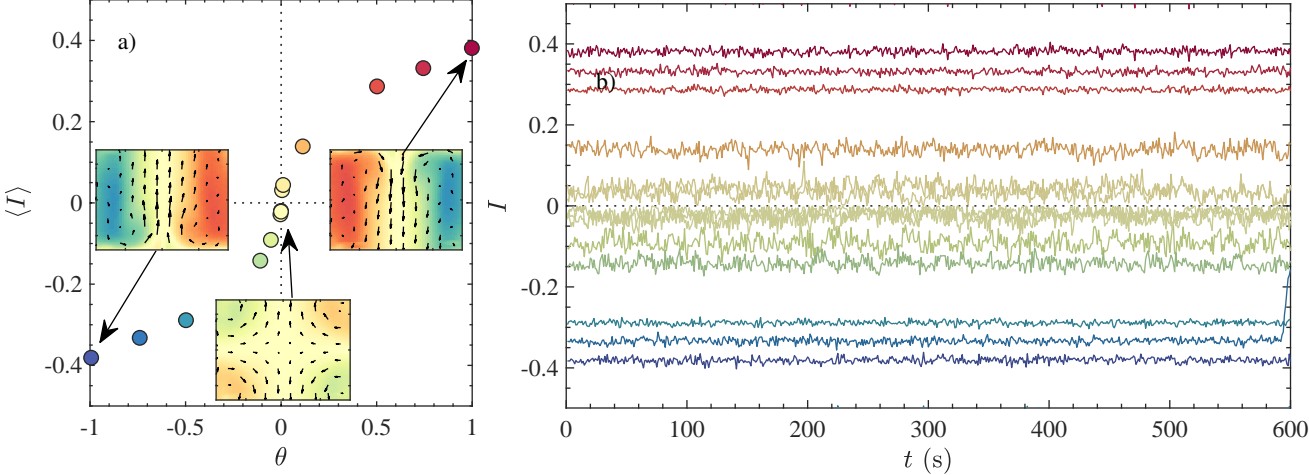

**Figure 4.** Seasonal cycle in the von Kármán experiment, imposed by modulating the vertical angular momentum imbalance $\theta$. a) Mean vertical angular momentum $\langle I \rangle$ response. b) Instantaneous vertical angular momentum response for a fixed $\theta$. The $z$ axis is oriented upward, providing the sign of the vertical angular momentum. The curves represents instantaneous values of $I$. Lines are colored according to $\theta$, as done in Panel a).

the heat flux imbalance between the poles and the Equator as the Earth revolves around the Sun. An example of the reaction of the mean vertical angular momentum $\langle I \rangle$ to such a seasonal cycle in the von Kármán water experiment is shown in Figure 4a. On average, the flow responds continuously – yet in a non-linear fashion – to the forcing imbalance $\theta$: the shift between the A and B states is progressive, as observed on Earth. Indeed boreal summer and austral summer circulations have different characteristics on our planet and they are often studied separately (Vrac et al., 2014). For a given value of $\theta$, the instantaneous value of $I$ is however fluctuating, as illustrated in Figure 4b, and similarly to the fluctuations of the daily temperatures on Earth.

### 4.3 Transitions induced by changes in the velocity fluctuations intensity

We can now apply a perturbation to our system by imposing higher velocity fluctuations ("increasing $CO_2$"), using blades that are more curved. In that case, we observe an interesting bifurcation: when the level of fluctuations is sufficiently high, the seasonal cycle becomes discontinuous and the transition between the A and B states becomes brutal, with a jump in the global mean vertical angular momentum $\langle I \rangle$, as seen in Figure 5. In addition, this discontinuous seasonal cycle shows hysteresis: once we have switched to one of the summer branches, if we decrease again $\theta$, the flow jumps directly to the other summer branch, without going through the T state (Ravelet et al., 2004; Ravelet, 2005). Once the flow is in one of the two summer states, its configuration remains the same permanently, unless the parameters of the forcing are changed. The transition from one of the summer states to the T state seems forbidden, and we have never been able to observe it.

The symmetric T state is then particulary difficult to reach, as it becomes marginally stable: starting from a symmetric state at $\theta \sim 0$, the flow switches to either of the summer states after a time that diverges as $\theta^{-6}$(Ravelet, 2005). Thus, the lifetime of the two-cell symmetric state is then drastically shortened even for small values of $|\theta|$. This observation is reminiscent of what happens on Earth. For the atmospheric circulation, recent studies suggest that increasing fluctuations and concentration of $CO_2$ disrupts the shoulder seasons dynamics. In Cassou and Cattiaux (2016), a clear example of this disruption of seasons is provided, whereas an analysis of the way in which altering greenhouse gases concentration disrupts the proportion of zonal *vs* blocked states of the atmospheric dynamics is provided in Faranda et al. (2019a).

### 4.4 Spontaneous jumps between circulation states under a fixed torque difference

Up to now, we focused on circulation changes induced by externally-imposed changes of impeller rotation rate imbalance $\theta$: these are *forced transitions*, similar to the ones caused by changes of solar radiation on Earth, resulting in seasonal cycles or glacial-interglacial transitions. But the von Kármán flow is actually even more interesting. Using another type of forcing, which is actually closer to the forcing conditions of the natural flows on Earth, we also observed interesting *spontaneous transitions* of the circulation (Marié, 2003; Ravelet, 2005; Saint-Michel et al., 2013). In this situation we set the torque imbalance applied by the impellers onto the fluid $\gamma = (c_1 - c_2)/(c_1 + c_2)$, imposing a constant *flux of vertical angular momentum* in the experiment. The new type of dynamics only happens when the level of fluctuations is sufficiently high. It is illustrated in Figure 6, where we observe spontaneous transitions between a summer state and the T state. The transitions can be quasi-periodic (case (a)) or very rare (case (b)), depending on the value of $\gamma$. They can be viewed as a laboratory equivalent of "weather regimes" (Vautard, 1990), of El Niño events, or of the Heinrich events of Figure 1.

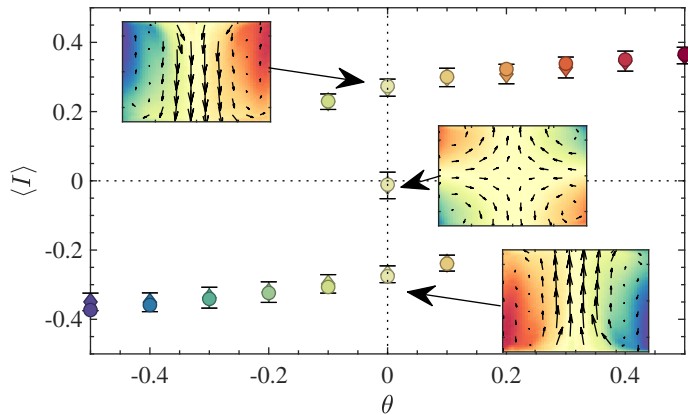

**Figure 5.** Seasonal cycle in the von Kármán experiment, for a case with high fluctuations. The response of the mean vertical angular momentum $\langle I \rangle$ to an asymmetry $\theta$ is now discontinuous. The spring/autumn branch is now reduced to the single central point here, disconnected from the austral and the boreal summer branches, which are respectively defined as the continuous branch extending to $\theta \to +1$ (respectively $\theta \to -1$), using notations consistent with Figures 3 and 4(a). Transitions are only possible from the spring branch to one of the summer states, or between the two summer states. Three distinct flow configurations are then possible for $-0.05 \leq \theta \leq 0.05$. Their topology is recalled in the insets.

These observations show that fast spontaneous transitions between long-lived states may arise in a complex system with many degrees of freedom and large fluctuations even when the external forcing does not vary as a function of time. Similar transitions could then occur in the atmospheric and oceanic circulation on Earth should the level of perturbations – our greenhouse gases emissions – become sufficiently high. Current models employ forcing functions for solar dynamics and CO2 concentration that possess a smooth structure, and bifurcations and sharp transitions in climate models are avoided. The von 260  Kármán analysis shows however that small scale fluctuations are important for the large scale dynamics, because they can trigger sudden transitions in macroscopic states.

## 4.5   The low-dimensional attractor

Despite their apparent complexity, the spontaneous transitions can actually be characterized by low-dimensional objects called attractors. This is illustrated in Figure 7, where we show the joint probability density function (PDF) of having simultaneously 265  a rotation frequency $f_1$ at the top and $f_2$ at the bottom impeller. Each subplot of Figure 7 corresponds to a different value of the torque imbalance $\gamma$. We see that in each case the joint PDF concentrates on a well-defined set, which we call the attractor. It can be a round blob, reflecting the existence of a fixed point in the dynamics [cases (a) and (f)] where the circulation remains either in the spring or summer state, or a more extended object, corresponding to transitions between the states. We have checked (Saint-Michel, 2013; Saint-Michel et al., 2013) that the transitions between states follow on average the same paths (delineated 270  by the colored arrows in sub-panels (c) and (d)) . Using tools from the dynamical systems community, we have been able to prove that all these observations were consistent with the existence of an attractor resulting from the coupling of the periodic

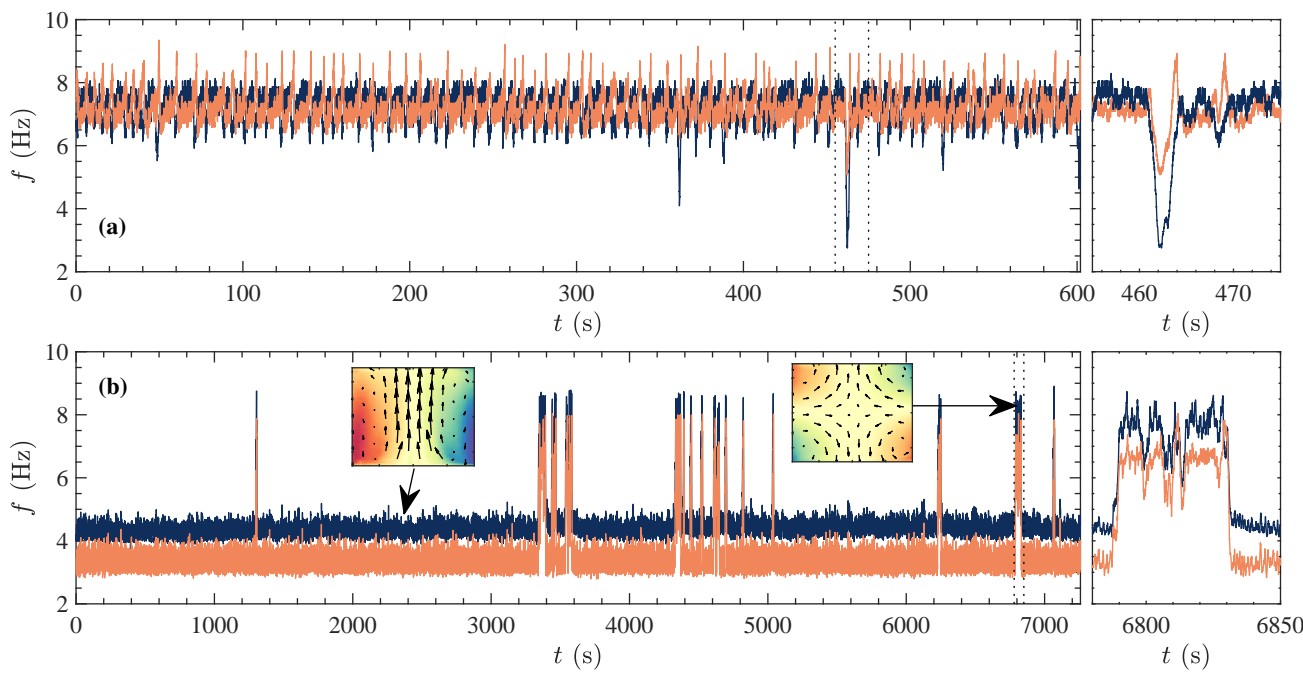

**Figure 6.** Spontaneous jumps between two circulation configurations observed by measuring the impeller rotation rates, for different, fixed, values of the torque imbalance $\gamma$. (a) Quasi-periodic, intermittent case, observed for small $\gamma$. (b) Rare events case, observed for a larger value of $\gamma$. The sudden transitions in the rotation frequencies are signatures of changes in the circulation shape. Note the different time scales. [Figures adapted from the PhD Thesis of B. Saint-Michel (Saint-Michel, 2013)]

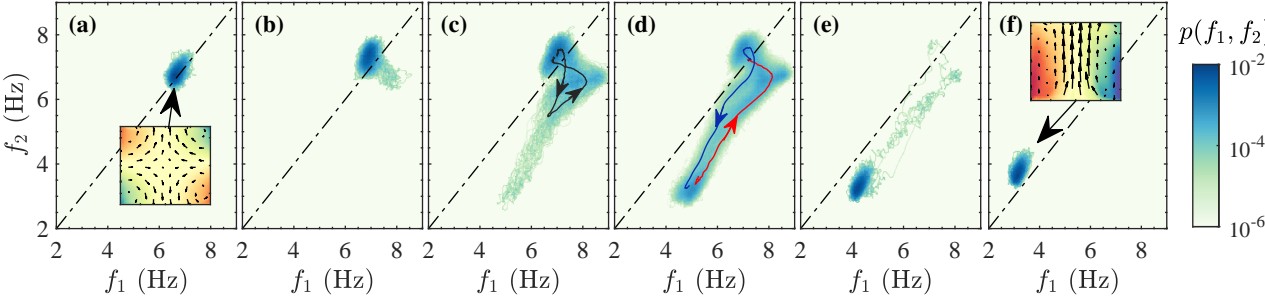

**Figure 7.** Attractors for different values of the torque imbalance $\gamma$. The colors code for the probability $p(f_1, f_2)$ that the top and bottom impellers rotate instantaneously at $f_1$ and $f_2$. They delineate a peculiar shape, that is supported by an attractor. As $\gamma$ is varied, we switch from a situation where spring is a fixed point [case (a)] to a case where summer is a fixed point [case (f)], with different intermediate situations, where the system can occasionally escape from its dominant state [cases (b) and (e)], or transition regularly between the two states, following a low dimensional attractor [cases (c) and (d)]. [Figures adapted from the PhD Thesis of B. Saint-Michel (Saint-Michel, 2013)].

forcing of the classical deterministic Duffing attractor (2D) with a Langevin equation (1D). Moreover, the attractor has fractal dimension between 3 and 10 (Faranda et al., 2017). This fractal dimension is very small compared to the number of degrees of freedom of the system $N \sim 10^{13}$. This study has motivated the development of a minimal model of the effective dynamics of the midlatitude jet stream (Faranda et al., 2019b). This model has been used to explore a range of possible behaviors beyond those displayed in the available data that could have appeared in past climates and could appear again in future climates. Similarly to the model derived by Faranda et al. (2017) for the von Kármán turbulent flow, the jet model is based on a coupled map lattice. Each element of the lattice reflects the dynamics of the jet at a given longitude. Stochastic forcing is used to simulate the alternance of cyclones and anticyclones, the geography and the small scale dynamics.

The existence of a low dimensional stochastic attractor in the experiment illustrates the fact that some degrees of freedom are probably not necessary to capture the bifurcations. It leaves the hope that maybe the essential features of the bifurcation will remain even if we decrease the number of degrees of freedom. So, let us see what happens when we reduce the number of degrees of freedom by increasing the viscosity, which is what is done in climate models when turbulent viscosity is introduced.

## 4.6 Circulation properties in glycerol

We have therefore conducted additional studies in the von Kármán cell filled with glycerol. This amounts to reducing the degrees of freedom to $\sim 10^4$, a number two orders of magnitude larger than the dimension of the stochastic attractor. In this reduced system, we observe again three different states, one symmetric with respect to the $\mathcal{R}_\pi$ symmetry, and two states that are exchanged by applying the $\mathcal{R}_\pi$ symmetry. They are shown in Figure 8. They have the same global topology than the equivalent states in the full system, even though they differ in details: for instance, the global rotation of the fluid is significantly more localized in the summer states, and only extends in the vicinity of the rotating impeller.

One can also drive a forced seasonal cycle in this configuration by varying $\theta$ and examining the associated variations of the vertical angular momentum $\langle I \rangle$, as shown in Figure 9-a. One sees that the seasonal cycle $\langle I \rangle(\theta)$ is continuous from winter to summer, similarly to the case of the full system with low fluctuations. The seasonal cycle also looks almost linear in $\theta$ and smaller differences in the vertical angular momentum $\langle I \rangle$ are seen between summer and winter.

The instantaneous global vertical angular momentum $I(t)$ (displayed in Figure 9b) shows very little fluctuations as a function of time regardless of the impeller frequency imbalance $\theta$: the fluctuations corresponding to the varying daily temperatures are suppressed. In this reduced system, we are thus able to capture *forced* circulation transitions, but not the short term fluctuations of the observable: in other words, we are able to conduct "climate" experiments (testing the switching between glacial and interglacial state), but not weather experiments (reproducing the daily temperature records) or amplitude of transition between summer and winter.

In a sense, this is quite an achievement: with a much smaller number of modes than the real system, we are able to forecast the dynamics of natural systems at a climatic level. Is this really so? Given that the number of degrees of freedom is still larger than the dimensions of the stochastic attractor, we may think that we should also be able to capture the bifurcations and the dynamics when switching to more curved blades, and imposing a torque imbalance.

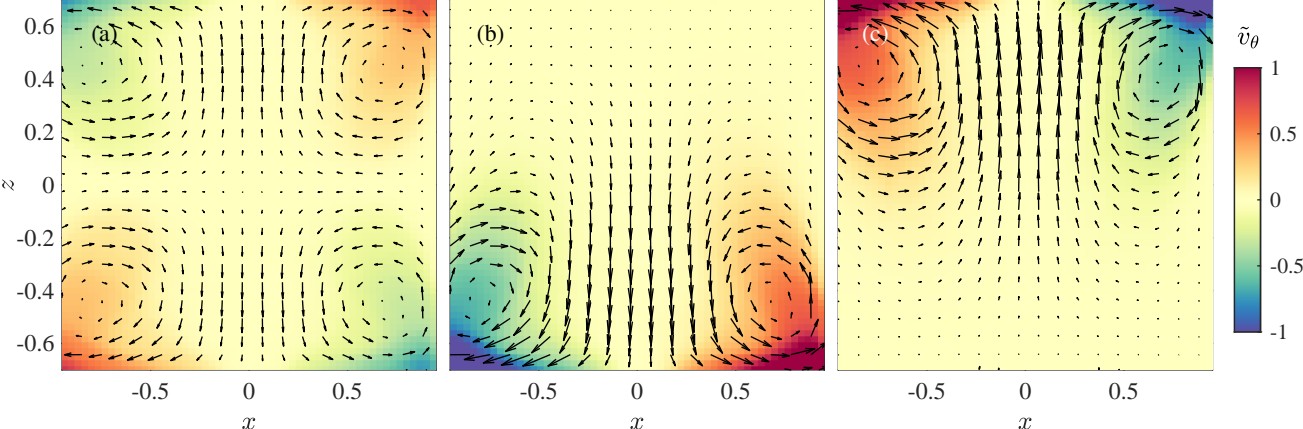

**Figure 8.** Different states observed in our reduced system – using glycerol – with fewer degrees of freedom: (a) Two-cell symmetric (T) state. (b) One-cell asymmetric (A) state. (c) One-cell asymmetric (B) state. The out-of-plane azimuthal velocity component $u_\phi$ is shown in color, while the in-plane radial and vertical speeds $(u_r, u_z)$ are represented by arrows. Fields have been decimated by a factor of 2 for better visibility. The positions $x$ and $z$ are normalized by the radius of the cylinder $R$.

This hope however got shattered by our experiments: at high viscosity, the multiple-branches region of Figure 5 disappears (Ravelet, 2005; Saint-Michel, 2013; Saint-Michel et al., 2013), along with the bifurcations and their associated dynamics, as shown in Figure 10. The contrast with the flow in water – the full system – is striking: not only does the stochasticity disappear (a reasonable observation if we believe that stochasticity is induced by the small scales), but so does the attractor itself! This means that our way of separating the large and small scales is not suitable for our system, as it has suppressed some of the degrees of freedom that are essential for the large scales dynamics. Instead, one has to pick up carefully chosen large, intermediate and small scales, to represent both the attractor, and the transitions between its branches. The question of how to do it is however still open and left for future work.

## 5 Fluctuations and small-scale properties

We have seen in the previous section that the small-scale velocity fluctuations are essential to get the rich transition dynamics between metastable states. The usual mental image invoked in such cases is that of an "energy" landscape (Fig. 11) in which the summer and spring states are represented by potential wells of different depths. Once the system is e.g. in the spring state (at position $x_-$), it can only jump towards the other potential well (at location $x_+$) following fluctuations exceeding the energy barrier $\Delta U_0$. This classical problem of stochastic processes results in an exponential distribution of escape times from the potential well, that we have observed experimentally (Ravelet, 2005). This kind of scenario has also recently been observed in stochastically forced PLASIM model of climate (Margazoglou et al., 2021).

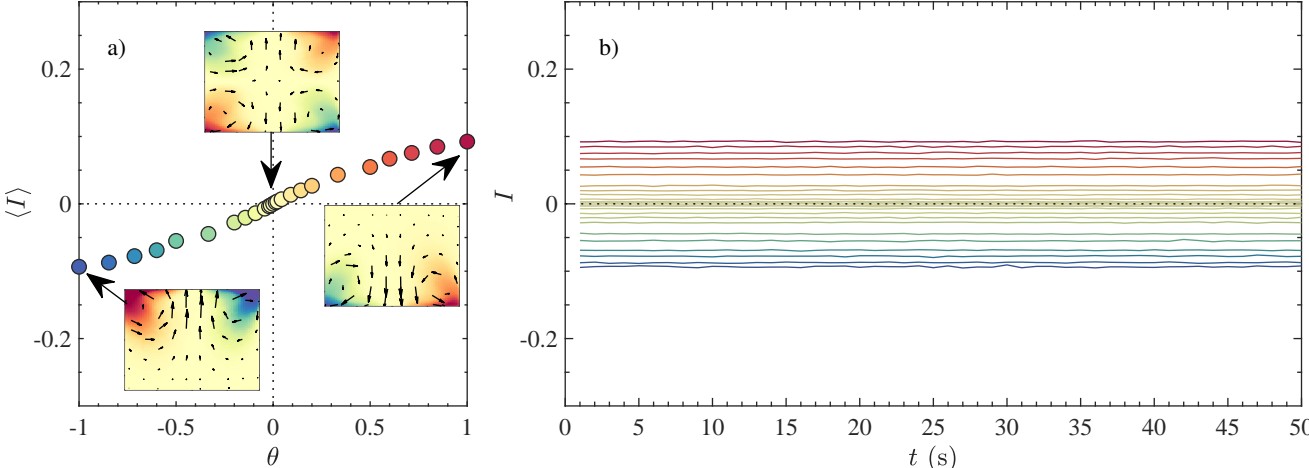

**Figure 9.** Seasonal cycle in the von Kármán experiment for a reduced number of degrees of freedom – using glycerol. a) Mean angular response $\langle I \rangle$ to the forcing asymmetry. b) Instantaneous vertical angular momentum response for a fixed $\theta$ colored according to $\theta$ with the color choices of Panel a).

How does one get fluctuations of sufficient amplitude to trigger these transitions ? Let us take a closer look at fluctuations in our von Kármán flow. They are shown in Figure 11a, from data issued from a direct numerical simulation of our experiment by H. Faller and collaborators, using the SFEMANS code (Cappanera et al., 2020). Large velocity fluctuations are primarily observed at two locations: one being close to the 'corner' of the experiment (i.e. close to the impellers and the outer wall), and the second being at mid-height. The first location obviously corresponds to the fluctuations induced by the moving boundary conditions which are then advected by the mean velocity of the spring state of Figure 3a. The second location is associated to a strong shear layer resulting from the counter rotation of the two main toroidal cell of the spring state. We then observe, in agreement with our physical intuition, that larger velocity gradients produce stronger velocity fluctuations.

What is the physical process that generates strong velocity gradients in a turbulent flow? Obviously, not the viscous dissipation, which has the opposite effect of smearing out such gradients. In fact, only the non-linear term of the Navier-Stokes equations $(\mathbf{u} \cdot \nabla)\mathbf{u}$ can produce finer length scales through triadic mode interactions. At a given scale, the intensity of the effect of the non-linear term can be estimated through the "local energy transfer" term defined as: $\Pi_\ell = \nabla_\ell \overline{(\delta_r u)^3}^\ell$, where $\delta_r u = u(x+r) - u(x)$ is the velocity increment over a distance $r$, and $\overline{x}^\ell$ indicates that we average the distance $r$ over a ball of size $\ell$ centered on $x$ . Indeed, this term provides the contribution of non-linear interactions to the energy budget, and provides the quantity of energy that is transferred through scales (Dubrulle, 2019). In other words, the larger $\Pi_\ell$, the higher the energy cascade towards smaller scales, and the stronger the final gradient (and velocity fluctuations) at this location. The quantity $\Pi_\ell$ is shown in Figure 12b): as expected, areas of large velocity fluctuations correspond to areas of large local energy transfers $\Pi_\ell$.

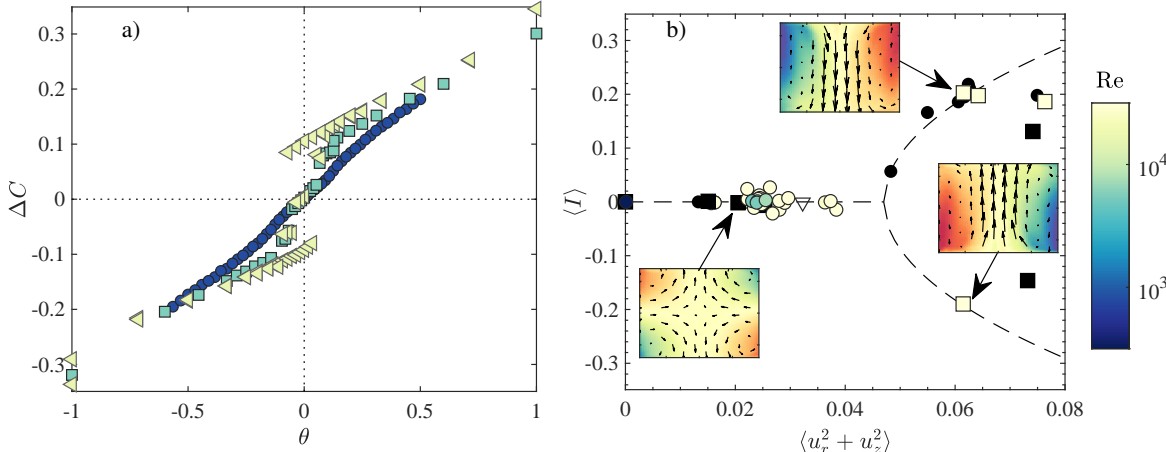

**Figure 10.** Disappearance of the bifurcation when the Reynolds number or the fluctuations are decreased. (a) Difference between the non-dimensional torque applied by the turbulence on the upper and lower impellers $\Delta c$ as a function of $\theta$, shown for various Reynolds numbers: blue circles: $Re = 980$; green squares: $Re \sim 5600$; yellow triangles: $Re = 11000$. The quantity $\Delta c$ is a proxy of $\langle I \rangle$. We see that the transition between the two summer states is discontinuous for $Re > 5600$ but becomes continuous at low Reynolds number. [Figure adapted from Ravelet (2005)]. (b) Linking the hysteresis cycle properties to the velocity fluctuations in the von Kármán flow following Thalabard et al. (2015). Colored symbols: Mean vertical angular momentum $\langle I \rangle$ as a function of the energy of azimuthal fluctuations $\langle u_\phi^2 \rangle - \langle u_\phi \rangle^2$ measured in our experiment at $\theta = 0$ for different impeller shapes and Reynolds numbers. The black symbols are $\Delta c$ at $\theta = 0$ at different Reynolds numbers and for different propellers shape. Above $\langle u_\phi^2 \rangle - \langle u_\phi \rangle^2 = 0.05$, two branches of the circulation exist; the lower branch corresponds to austral summer, while the upper branch corresponds to boreal summer. The spring/autumn branch is unstable. For $\langle u_\phi^2 \rangle - \langle u_\phi \rangle^2 < 0.05$, the austral (and boreal) summer branches disappear and we observe only one possible state of the circulation corresponding to the spring/autumn branch.[Picture adapted from Thalabard et al. (2015)].

## 5.1 Scaling exponents and intermittency

Can we quantify the connection between strong velocity fluctuations and large energy transfer events in a more rigorous way? To do so, we can introduce a local diagnostic quantity that will prove useful in the understanding of the local dynamics of the energy transfer. It is the "local scaling exponent", defined as $h = \ln(|\delta_r u|)/\ln(r)$, or equivalently, $|\delta_r u| \sim r^h$. This quantity is connected to the mathematical notion of Hölder continuity, which provides a weaker regularity condition than differentiability. A given velocity field is said Hölder continuous with some exponent $h < 1$ (i.e. not necessarily differentiable) at small scales if the following holds:

$$|\mathbf{u}(\mathbf{x} + \mathbf{r}) - \mathbf{u}(\mathbf{x})| < Cr^h. \tag{1}$$

We see that Hölder continuity uses the velocity increment $\delta_r u$ as a building block. This is interesting, as a classical result by Kolmogorov states that under stationarity and homogeneity condition, the mean energy flux of a solution of Navier-Stokes

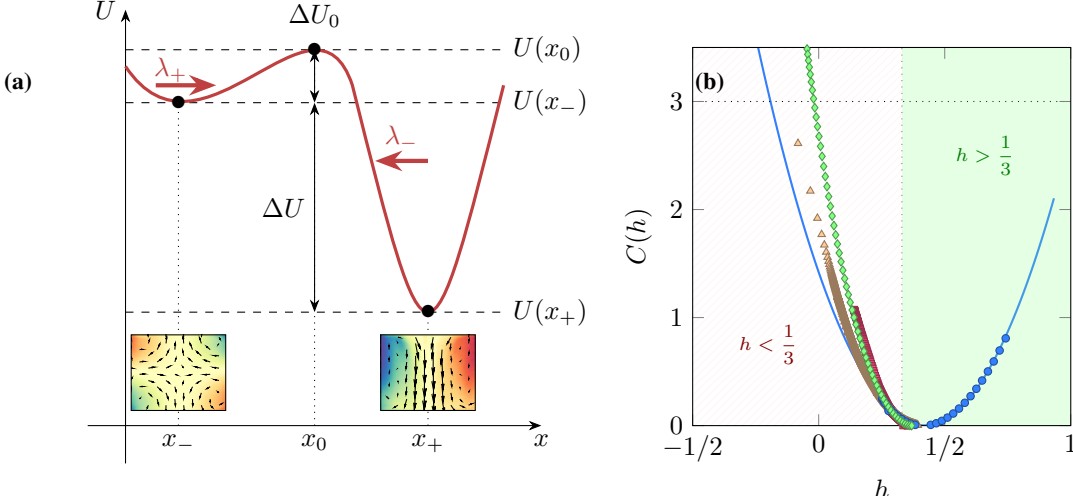

**Figure 11.** (a) Energy landscape picture of the bifurcation. The von Kármán flow evolves, as would a Brownian particle, between two potential wells of different depths $U(x_-)$ and $U(x_+)$ separated by energy barriers $\Delta U_0$ or $\Delta U + \Delta U_0$ depending on the nature of the transition. (b) Multifractal spectrum exponent $C(h)$ of the wavelet velocity increments ( equivalent to $|\delta_\ell u|$) computed using Equation (3) on numerical simulation data of the von Kármán flow obtained by Hugues Faller. The data points are given by blue circles, the blue line is a parabolic fit extrapolating towards $C(h) = 3$ which cannot be accessed with our limited data set. For comparison, we have also put the multifractal spectrum exponent computed form the scaling properties of the local energy transfer (green diamonds) in the same numerical simulation. Finally, we have add to the graph multifractal exponents derived from scaling exponents measured by Iyer et al. (2020) through the moments of the signed longitudinal (red squares) and transverse (yellow triangle) velocity increments. [Figure adapted from Faller et al. (2021)].

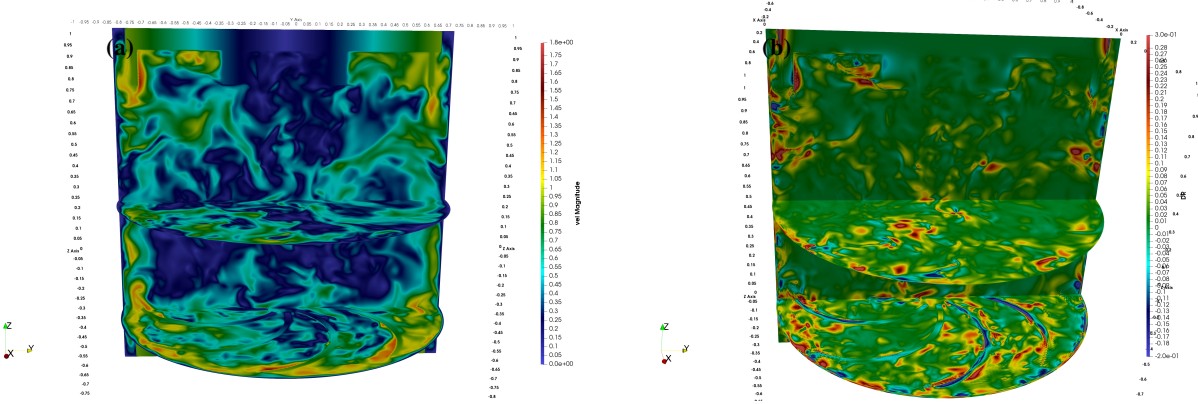

**Figure 12.** (a) Instantaneous magnitude of velocity fluctuations. (b) Local energy transfers in the von Kármán flow. [Pictures courtesy Hugues Faller].

equations is constant in the inertial range :

$$< \Pi_\ell > = -\epsilon, \tag{2}$$

where $\epsilon$ is the global energy dissipation. Integrating this over scale, one gets $< (\delta\mathbf{u}(\mathbf{x},\ell))^3 > = -(4/3)\epsilon\ell$ for $\ell$ in the inertial range. This is called Kolmogorov 4/3 law. If signed and unsigned (with absolute values) values of velocity increments were equivalent, and the turbulence were fully homogeneous, this would suggest that $h \sim 1/3$ for scales in the inertial range: turbulence would be a monofractal, with exponent $h = 1/3$.

However, these two hypotheses are violated. First, it is clear that unsigned moments converge much more rapidly and easily that signed moments, as the latter are prone to cancellations effect, and sensitive to large fluctuations of positive or negative value. Due to this phenomena, signed moment are sensitive to possible sub-leading correction to scaling, that are avoided by taking the absolute value. In that sense, the local scaling exponent we define is more robust, but maybe less sensitive to subtle scaling effects, like oscillatory scaling behavior. In the von Kármán flow, we have checked that indeed $< (\delta\mathbf{u}(\mathbf{x},\ell))^3 > \sim \ell$, while $< |\delta\mathbf{u}(\mathbf{x},\ell)|^3 > \sim \ell^{0.8}$ (Faller et al., 2021). Second, we have observed that at small scale, the turbulence is strongly inhomogeneous (Faller et al., 2021), so that it is not clear that a single exponent can be sufficient to describe the flow.

While we cannot consider the 4/3 law as an exact constraint pointing to $h = 1/3$, we can still find a relic of such value using the scale dynamics of turbulence. Indeed, from dimensional analysis, it is possible to show that the local energy transfer scales like $\Pi_\ell \sim \ell^{3h-1}$ (Dubrulle, 2019) (see however Schertzer and Lovejoy (2011) for multifractal transitions allowing deviations from such dimensional scaling). So, when $h > 1/3$, $\Pi_\ell$ decreases with scale and tends to zero, while for $h < 1/3$, $\Pi_\ell$ increases with decreasing scales, and is able to provide larger and larger fluctuations. The case $h = 1/3$ is special, and corresponds to a case where the local energy flux is scale invariant. This is the situation that was assumed in the original self-similar theory of turbulence proposed by Kolmogorov. This theory leads to an energy spectrum scaling as $k^{-5/3}$, yet we see from Figure 12b that the local energy transfer term is highly variable in space, contrasting with the assumptions of homogeneity and isotropy made by Kolmogorov. This means that we have to forget about global homogeneity or scale invariance for turbulent flows, which would provide us with a single value for the scaling exponent, and rather consider the possibility that there may be a whole bunch of interesting local scaling exponents. To capture them, we must find a way to describe them using local scaling laws.

## 5.2 The multifractal spectrum

The corresponding mathematical description can be built using the large deviation theory (Eyink, 2007-2008) that allows us to describe the probability of finding an exponent $h$ at a given place or time in the flow as

$$P\left\{|\ln(\delta_\ell u)| = h\ln(\ell/L)\right\} \propto \exp\left[\ln\left(\frac{\ell}{L}\right) C(h)\right] = \left(\frac{\ell}{L}\right)^{C(h)} \tag{3}$$

where $C(h)$ is the rate function of $h$, also called multifractal spectrum. Originally, $C(h)$ has been interpreted as the statistical codimension of the set where the velocity increment at a distance $\ell$ scales like $\ell^h$ (see Frisch and Parisi (1985); Schertzer and Lovejoy (2011) and references therein) . In that interpretation, it is natural than $C(h)$ be less than the space dimension

($C(h) \le 3$). In the large deviation interpretation, however, one can get arbitrarily large $C(h)$, corresponding to arbitrarily rare events. The codimension interpretation nevertheless allows to single out the exponents where $C(h) > 3$ (possibly $C(h) \to \infty$), meaning that the corresponding events are rarer than point-distributed events, which indeed corresponds to exponents with a very low probability of occurrence. More generally, the large deviation property enables to make a direct connection between the multifractal spectrum, and the scaling exponents of the $n^{th}$ moment of $|\delta_\ell u|$, as :

$$< |\delta_\ell u|^n > \quad \sim \quad \ell^{\zeta(n)},$$
$$\zeta(n) \quad = \quad min_h(nh + C(h)). \tag{4}$$

This is a Legendre transform. In that description, the value of $h$ at which $C(h)$ is minimum corresponds to the exponent with highest probability, $h_{mp}$. This exponent is usually very easy to measure, as it corresponds to an exponent that is frequently hit.

   We have computed the multifractal spectrum in the von Kármán flow, using both experimental measurements and numerical

simulations. The result is shown in Figure 11b. We see that the minimum of $C(h)$ is attained for $h_{mp} = 0.35$, very close but slightly larger than the $1/3$ Kolmogorov exponent. The fact that it is not exactly equal to $1/3$ is a hallmark of intermittency, as can be understood by coming back to the $4/3$ law. This law is strictly valid only for the signed exponent, but let us assume for the moment that it gives a good approximation for our absolute value moment, meaning $\zeta(3) = 1$. Given the Legendre property (4), this imposes that we must have $C(h) \le 1 - 3h$, which means that at $h = 1/3$, $C(h)$ is positive. In the codimension

interpretation, this means that $h = 1/3$ is true everywhere but for an ensemble of dimension $3 - C(h)$. The only way it can be true everywhere is if $C(1/3) = 0$, which means that $C(h)$ is a delta function centered on $1/3$ (a pure fractal). Indeed, consider a simple model in which the multifractal spectrum is parabolic around its minimum $C(h) = (h - h_{mp})^2/2b$. This is usually true near a minimum, and assuming that it extends further away was first done by Kolmogorov (1962), resulting in the log-normal model of turbulence. If we now impose that $\zeta(3) = 1$ in this model, we then get $h_{mp} = 1/3 + 3b/2$, so that the only

free parameter is $b$, the scale parameter. Therefore, in this model, the only value for which $h_{mp} = 1/3$ is $b = 0$, corresponding to a delta function for $C(h)$, and then, to a pure fractal.

   If this simple model were valid in the von Kármán flow, our measured value of $h_{mp}$, would correspond to $b = 0.011$, meaning a rather mild intermittency. Given that the condition $\zeta(3) = 1$ does not apply exactly for the absolute moment in our case, a more precise measure of intermittency can be made by performing a parabolic fit to our data, resulting in a larger value

$b \sim 0.035 \pm 0.01$. We stress here that the intermittency property is not in contradiction with the flux conservation of eq. (2) : we have indeed checked that $< \Pi_\ell >$ is scale independent in the inertial range, and equal to the global energy dissipation. Locally, there is however no reason for the local flux to be scale independent, as it can hit regions with $h \ne 1/3$, with non zero probability (intermittency).

   At all the places where $h < 1/3$ (and we see that there are quite a few), the local energy transfer increases with decreasing

scales. This means that there are many places in a turbulent flow where large fluctuations can build up. The smallest exponent we were able to measure is $h \approx 0.17$. We may also compute the multifractal spectrum using the scaling laws of the local energy

transfers:

$$
\begin{aligned}
<|\Pi_\ell|^n> &\sim \ell^{\tau(n)}, \\
\tau(n) &\sim min_h(n(3h-1)+C(h)).
\end{aligned}
\tag{5}
$$

Using the local energy transfers, we are actually able to observe a smaller local scaling exponent, very close to $0$. These values can be compared to values obtained by computing the multifractal spectrum via application of the Legendre transform to scaling exponents computed in recent high Reynolds number numerical simulations of Iyer et al. (2020), for both longitudinal $\delta u^L = \delta u \cdot \ell/\ell^2$ and transverse $\delta u^T = \delta u \times \ell/\ell^2$ velocity increments. One see that the longitudinal velocity increments also provide a minimal value around $h \approx 0.16$, while the minimal value for transverse velocity increments is $h \approx 0$. Smallest exponents, if they exist, will become increasingly rare, and we need probably to wait order of magnitude longer time to be able to have a chance to observe them. To get an idea of what could be the smallest exponent we may observe, we can extrapolate our measurements with the log-normal model. The parabolic extrapolation to the value where $C(h) = 3$ shows that the smallest exponent we may be able to find in the flow with a non-zero probability is $h_{\min} \simeq -0.2$. For such a negative value of $h$, the local energy transfer increases sharply towards the smaller scales like $\Pi_\ell \sim \ell^{-1.6}$. Does it increase without limit? Is there any process to stop it?

## 5.3 The regularizing scale

Indeed, at the smallest scales of the flow, viscous effects curb the growth of the local energy transfer. The typical scale below which viscous effects become effective can be estimated from the dimensional analysis arguments of Paladin and Vulpiani (1987). Their argument is that the local energy budget of a turbulent flows at a given scale $\ell$ includes a sink term due to viscous dissipation besides the local energy transfer we mentioned in the previous sections (for a complete expression of the energy budget, see Dubrulle (2019)).

The sink term reads $\Pi_\nu = \nu \overline{(\delta_r u)^2}^\ell$ and scales like $\Pi_\nu^\ell \sim \nu \ell^{2h-2}$ at locations where $\delta_r u \sim r^h$. The viscous term therefore balances the local energy transfer term at a scale $\eta_h \sim \nu^{1/(1+h)}$. This scale thus appears as a fluctuating cut-off scale, which depends on the scaling exponent and therefore on $\mathbf{x}$. This is a generalization of the Kolmogorov scale $\eta_K \sim \nu^{3/4}$. Below $\eta_h$, the sink term due to dissipation becomes dominant in the energy budget, the local energy transfer vanishes and the flow becomes regular, whereas this regularization occurs everywhere at $\eta_K$ in the K41 model.

On Earth, strong velocity gradients develop in regions associated with Planetary Boundary layer and may result in large scale extreme weather events: tornadoes, hurricanes or supercells. Besides viscosity, there are other types of regularizing mechanisms for such phenomena. The energy can indeed be dissipated directly on solid surfaces with sometimes dramatic consequences for human beings, animals and vegetation. Such a process is however akin to a "large scale friction" that is not present in our experiment. Extreme weather phenomena are still difficult to forecast. To capture them, is it enough to refine our every day weather forecasts at a resolution of $\sim 1$ km (the scale of convective flows) (Hohenegger et al., 2015)? This is the topic of the next Section.

## 5.4 The computational nightmare continues!

We have seen in Section 4 that we need to include the velocity fluctuations of the von Kármán turbulent flow down to their smallest scales to accurately reflect the large-scale properties and dynamics of its seasonal cycle. For any given local exponent $h < 1/3$, the local energy transfer increases with decreasing scales down to $\ell = \eta_h$. Therefore, we have to make sure that our modeling resolution $\delta$ (the size of our grid) is below $\eta_h$ for any $h$ present in the flow. Otherwise, we may underestimate the magnitude of the fluctuations by a factor $(\delta\eta_h)^h$. A stringent criterion to make sure all small scales are properly taken into account would be to choose $\delta$ equal to $\eta_{h=h_{min}}$ where $h_{min}$ is either the smallest measured exponent **observed** in our simulation (optimistic view) or the smallest **estimated** exponent, computed using , e.g. parabolic extrapolation (pessimistic view). In the first case, this gives $h_{min} \sim 0$, resulting in $\eta_{h=h_{min}} \sim 1/\text{Re}$, smaller than the Kolmogorov scale by a factor $Re^{-1/4}$ . In the pessimistic view, $h_{min} \sim -0.2$, resulting in $\eta_{h=h_{min}} \sim \text{Re}^{-5/4} \sim \eta_{\text{K}}\text{Re}^{-1/2}$. This pessimistic choice actually increases the number of degrees of freedom required to accurately model turbulent flows by a factor $\text{Re}^{3/2}$ compared to our initial estimate based on the Newtonian viscosity of the fluid and the K41 turbulent model. We then reach $N \sim 10^{23}$ degrees of freedom (the Avogadro number) in our water von Kármán experiment, or $N \sim 10^{40}$ for the atmosphere!

## 6 Conclusion

We seem to have understood why climate models work in the first place: the large scale topology and externally-forced transitions do not depend very much on the value of the viscosity, and may be in fact described with tools from statistical mechanics (Thalabard et al., 2015), involving only a few hundred to thousands modes that can easily be captured by present climate models. However, when it comes to understanding whether present climate model have enough degrees of freedom to capture global bifurcations, we have reached contrasted conclusions using our laboratory experiment.

A key result from this experiment is that the large scales of even highly turbulent flows are not restricted to have purely relaxational dynamics, decaying monotonically to an "energy landscape pit" from which escape is impossible, the turbulent fluctuations "randomizing away" any escape attempt. The forced rotation rate experiments (section 4.3) show that this is definitely not the case: when brought to the symmetric forcing $\theta = 0$ situation, the flow remembers with seemingly infinite memory how it has been prepared, and remains in its "A" or "B" state indefinitely (the "T" state is peculiar in that its decay time diverges when $\theta \to 0$, but the transition seems *a priori* possible). In this respect, though the level of turbulent fluctuations is very high, the large scales of the flow are as deterministic as a light switch. This behaviour is in fact well known in aerodynamics, where it is encountered in the "stalling" transition of airfoils (Sarraf et al., 2005). The fixed flux experiments (section 4.4) are even more spectacular, in that they show that the large scales of the flow can have non-trivial temporal dynamics, with an elaborate scenery of fast transitions between long-lived states, with residence times in the states either sharply distributed around a finite value, or exponentially distributed with a long characteristic time of decay.

The fact that this time-evolution of the large scale topology of the flow can be described by a low dimensional attractor, corresponding to few degrees of freedom, is very encouraging, and justifies the search for procedures for cutting down the number of degrees of freedom. However, the destruction of the coherent large-scale dynamics by the large increase of the

viscosity in the glycerol experiments shows that a too trivial procedure might preclude a sufficient representation of potentially vitally important large-scale Earth system processes. Indeed our study suggest that a good practice to approach complexity is to use both simple conceptual models such as the one presented in Faranda et al. (2017, 2019b) as well as complex models (e.g. DNS or LES for the von Kármán flow and full numerical integration of primitive equations for the climate). While simple models can quickly provide an overview of the bifurcation landscape of a system, fine models can be used to explore targeted regions of the phase space where interesting phenomena produce e.g., extreme dissipation events in the von Kármán flow or convective events in climate simulations.

We have seen that these fluctuations are generated at very small scales, by a concentration of local energy transfer ending up into point-like quasi singularities, with large fluctuations over small scales. These small scales are really small, even smaller than the Kolmogorov scale $\eta_K$. Resolving these small scales using direct numerical simulations comes at a colossal computational cost, out of reach of current (and future) computing facilities. So the problem of small scale parametrization is in fact even more acute than we thought. From what we have learned, one thing we must take care of is to devise a model that is able to reproduce and include the large velocity fluctuations observed at very small scale in experimental turbulent flows without explicitly resolving them. This is clearly the next challenge in climate modeling.

*Author contributions.* All the authors wrote the manuscript based on results obtained in collaboration with all the people cited in the acknowledgements

*Competing interests.* The author declare no competing interest.

*Disclaimer.* None.

*Acknowledgements.* Even though Bérengère Dubrulle received the medal as an individual, most of this work has been done through and thanks to collaboration with students, post-doc and colleagues, some of them being co-author of this manuscript. Didier Paillard co-supervised with Bérengère on topics related to the subjects and fueled our thinking with many interesting references and ideas. She learned a lot about climate and dynamical systems from discussion with Pascal Yiou, Robert Vautard, Didier Roche and other members of their team.

Thanks to Jean-Philippe Laval, Vishwanath Shukla, Florian Nguyen, Hugues Falller, Caroline Nore, Jean-Luc Guermond and Loïc Capanera for providing the numerical data and analysis. François Daviaud, Arnaud Chiffaudel, Adam Cheminet, Jean-Marc Foucaut, Christophe Cuvier, Yashar Ostovan, Vincent Padilla, Cécile Wiertel, Pantxo Diribarne, Pierre-Philippe Cortet, Éric Herbert, Davide Faranda, Ewe-Wei Saw, Valentina Valori, Romain Monchaux, Brice Saint-Michel, Simon Thalabard, Denis Kuzzay, Paul Debue, Damien Geneste performed the particle velocimetry measurements and analysis. The bifurcation measurements in water and glycerol owe much to the PhD work of Louis

Marié, Florent Ravelet, and Brice Saint-Michel. This work has been supported by the ANR EXPLOIT, grant agreement no. ANR-16-CE06-0006-01.

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
