# Peer review of "How many modes are needed to predict climate bifurcations? : Lessons from an experiment"

_Nonlinear Processes in Geophysics, 2021_

## Referee Comment (RC1)

This study concerns the modeling, through interpretation of laboratory experiments, of the dynamics of the very large features of the Earth system on climate time-scales. The manuscript (presumably) covers the Lecture for the Lewis Fry Richardson (EGU) medal of Berengère Dubrulle, with several of her co-authors. It is entitled *How many modes are needed to predict climate bifurcations? Lessons from an experiment.* The paper is placed in the broad framework of the question of whether a scientifically acceptable answer can be given to society concerning the evolution of climate, and what effort (theoretical, modeling, experimental and numerical) is needed in order to be able to do so reliably. Two examples are specifically examined: the switching in the atmosphere between convection and blocking, and the Niño–Niña transition, a prediction of which would be extremely useful to society (and to modelers of course). The principal tool is the experimental facility used by this group in Saclay (France), together with sophisticated mathematical tools. Specifically, the study is focusing on spontaneous transitions of the von Kàrman flow between two counter-rotating cylinders under different types of forcing. Reynolds numbers are of the order of $10^{13}$ (or less, when using glycerol instead of water), basically too small compared with geophysical flows, but still very enlightening. In the last sentence of the abstract and in the conclusion, the authors *stress the importance of describing very small scales to capture fluctuations of correct intensity and scale.* Of course, one thinks of the 50-year old Donnella Meadows *et al.* project at MIT that predicted a collapse of the world climate and economic system in or around 2020 with a model having a small number of relevant variables and different complex (feedback) interactions. Are these models enough? With such drastic truncations, how are the all-essential small-scales treated or incorporated?

One could argue that the equivalence between Earth systems and the von Kàrmàn (VK) flow is not quite as much as stated in the paper; in particular, what of the role of stratification, which is strong at large scale? What is the effect of shear? Can it be quantified? Do they play any role in these transitions? What about chemistry, moisture and the like?

Also, concerning the computational nightmare, sometimes called the curse of dimensionality, there are several papers claiming that neural networks can lead to avoiding that curse. Do the authors have a comment on that?

This paper is very pleasant to read. It describes the thinking going-on at the intersection of several fields of physics and the creativity in the development of new data and in interpreting it. I recommend publication.

Details:

1. In the very first sentence, it is a bit odd to see the first author (BD) talk of herself in the third person, ...

2. In the title, *fry* and *?:* might have to be changed

3. I have several problems with Figure 1 (l. 55+). What is a Dansgaard-Oeschger event? What is *ka*? What are M12345? What do the 8 12 14 18 etc refer to? Also, the insets are way too small, I suggest making the figure substantially larger. A reference (to Didier Paillard or other) would be necessary as well.

4. line 92+, perhaps one could cite and discuss a bit the drastic reduction of modes (down to 3) done by Ed Lorenz and which led to many investigations of lack of predictability of atmospheric flows. For example, why is 3 enough (when, as pointed out by the authors, we need a priori $10^{24}$ modes)? Why not 5 or 13? And what do larger systems bring compared to the small (and successful) system? When do we know that we have *enough* small scales? What criteria can be applied?

5. Figure 2: give the unit of sizes (0.925, 1, ...)

6. l. 164, ITCZ is not defined

7. Figure 3: is there a significance in the fact that the amplitudes in flow (a) are smaller than for (b) and (c)?

8. Figure 5: Spring and Summer states are undefined here. One could at least refer to Figure 3, or perhaps add a line in Table 1 defining the 3 (or more) states (Spring or autumn, Austral and Boreal Summers, and Transitional) since they are key ingredients of the analysis and the paper refers to them often.

9. Figure 11: Multifractal spectrum of what?

10. l. 345: Are there not strong velocity gradients in the Planetary Boundary Layer as well as the extreme events mentioned here? Do they not translate into large-scale extremes?

11. l. 351 and beyond:

12. l. 365: By *few modes*, do you mean 3 as in the Lorenz model, or $10^4$ as in the glycerol experiment described in the paper? Again, how little can it be? How does one determine this threshold? Is not the *computational nightmare* mentioned in Section 5.4, *also* an experimental nightmare?

---

## Referee Comment (RC2)

**A review of "Lewis Fry Richardson Medal Lecture - How many modes are needed to predict climate bifurcations? : Lessons from an experiment" by Dubrulle et al., NPG-2021-19**

**General comment**

This manuscript focuses on a controversial and challenging question: *what can we learn about climate dynamics from a laboratory fluid experiment?*, and it claims that there is a lot to learn! This optimistic answer is argued using a given generality of dynamical systems, more precisely stochastic attractors, and turbulence phenomenology, namely cascades and finally multifractal intermittency. Moreover, this approach is presented in a nice and elegant way. It certainly helps readers to emerge smoothly within the text, despite some sharp theoretical corners that remain beyond the present paper considerations.

The paper is presented as a review, which is basically true as far as the work of the team and its laboratories is concerned. However, to my opinion, it would gain from taking into account the earlier findings of other teams although they did not start from a laboratory experiment, but, for example, from the stochastic interactions between weather and climate, both analysed in a multifractal way.

A certain number of clarifications seem necessary, in particular on the pivotal issue of the energy flux conservation. A subsequent question is whether all the conclusions would hold in the absence of energy flux conservation. Below is a series of detailed comments and some suggestions which will hopefully be useful for a revision of this challenging manuscript.

**Detailed comments and suggestions**

**Title**

It is surprising to see « Lewis Fry Richardson Medal Lecture » in the title, at least because the lecture was given by the Medalist, whereas the paper is cosigned by 4 other persons (see suggestions for L16-28 below).

**Affiliations**

Affiliations 1 and 2 are identical.

**L16-28:**

This paragraph should be rewritten slightly to make it clear that this article is based on the 2021 Lewis Fry Richardson Medal Lecture presented by B. Dubrulle (avoiding to put it in the title) and that both are the result of personal intuitions and collective work.

**L75-78:**

The current "astronomical number" of degrees of freedom is obtained in a 3D isotropic framework, while even larger number of degrees of freedom ($N \approx 10^{27}$) has been obtained in an anisotropic framework (e.g., Schertzer and Lovejoy, 1991), *a priori* much more relevant for the atmosphere.

**L89-93:**

It should be mentioned that the eddy viscosity has been known since at least Richardson (1926) to be highly scale-dependent, so the quoted estimates are only relevant at given scales that need be

specified. Moreover, the action of small scales on larger ones is not limited to eddy viscosity with the presence of a beating / backscatter / renormalized forcing term (e.g., Forster et al, 1977; Frisch et al., 1980; Herring et al. 1982, and references therein).

The introduction of an eddy viscosity is far from being a "procedure [which] may sound foolish and bound to failure", whereas it corresponds to a not so sophisticated renormalisation of the Green function that is indispensable to prevent accumulation of energy at the smallest explicit scales and the resulting explosion of the model (see references above).

However, it is far from being sophisticated enough, particularly with regard to the necessity to also renormalise forcing and intermittency. So there is no reason to claim that the resulting simulations are that close to reality. In particular, without providing the objective metrics that have been developed to measure it. (Minor note: the XXX surrounding the reference Flato et al., 2013 must be suppressed).

**Fig2 caption and other places**
It would be important to clarify that what is called "angular momentum" is *in fact* its projection on the vertical axis oriented from bottom to top.

**L145-152**
It might be useful to give more specific names to $O(2)$ and $SO(2)$, e.g. "(general) orthogonal group" and "special orthogonal group" respectively, instead of just the generic name "symmetry group" for both, as well as to recall that $SO(2)$ is the (connected) component of $O(2)$ whose elements have +1 determinants, the other component having determinants -1. This can be particularly useful in understanding the reduction of $O(2)$ into $SO(2)$ with the breakdown of rotational symmetry $\mathscr{R}_\pi$.

**L242**
It would be useful to make the mentioned stochastic Duffing attractor much less mysterious, for exemple, that it results from the coupling of the periodic forcing of the classical deterministic Duffing attractor (2D) with a Langevin equation (1D).

**L323-325**
The figure reference should point to Fig.11b, not Fig.12b.

$C(h)$ is rather the (statistical) co-dimension of the support of the singularity $h$ than a multifractal spectrum (Halsey et al., 1986), which originally denoted the (geometrical) dimension $f(h)= 3-C(h)$ introduced by Parisi and Frisch (1985), where $C(h)$ is then constrained to be $\leq 3$ (in a 3D embedding space). On the relations between dimension and co-dimension formalisms see, for instance, Schertzer and Lovejoy (2011) and references therein. In particular, singularities $h$ such that $C(h) > 3$ are almost surely not observable on a unique sample (Hubert et al. 1973).

The fact that the minimum of $C(h)$ occurs at $h_0 = 0.35 \approx 1/3$ is rather related to a weak intermittency of the mean energy flux, more precisely to a low co-dimension $C_1$ of the later: $C_1 = 3h_0 -1 = 0.05$ with both the assumptions of log-normality and conservation of the energy flux, i.e., the strict scale invariance of the mean energy flux density $\varepsilon_\ell \approx \delta_\ell u^3 / \ell \approx \ell^{-\gamma}$; $\gamma = 1 - 3h$. However, this estimate strongly disagrees with the other estimate $C_1 = 1$ obtained from $C(0)=1$, which belongs to the empirical parabolic fit $C(h)$ plotted in Fig.11b. This is because this adjustment supports the first assumption, but not the second.

Indeed, when obtained from the co-dimension $c_\varepsilon(\gamma)$ of a conservative flux: $c_\varepsilon(\gamma) = (\gamma + C_1)^2 / 4C_1$, the function $C(h)$ depends on the unique parameter $C_1$, whereas here it requires two independent parameters (horizontal shift and rescaling) to obtain a vertically orientated parabola. Although there is no obvious theoretical reason compelling that the points (0.35, 0) and (0,1) belong to the curve

$c(h)$, this seems to be supported by the empirical extrapolation. Furthermore, the empirical fit is precisely in agreement with the conservation of the statistical moment of order $q_1 \approx 0.18$ instead of $q = 1$ and all statistical moments of order $q > q_1$ diverge with increasing resolution, including the mean flux ($q = 1$). The latter corresponds to the estimate $h \approx 0.17$ given by the authors as the smallest measurable singularity.

I think the current failure to conserve the energy flux needs to be stated clearly and whether it is a system or a model failure needs to be clarified. A similar clarification should be brought on the impossibility to measure singularities $h$ lower/ $\gamma$ higher than those of the mean flux, as well as on the nature of this impossibility.

**L328-333**

It is important to distinguish the energy flux $\Pi_\ell$ from its density $\varepsilon_\ell$ (i.e. it is not just a terminology issue): $\Pi_\ell$ results from the 3D integration on a volume $\ell^3$ of its density $\varepsilon_\eta$ at much smaller scales $\eta << \ell$. Strong fluctuations of $\varepsilon_\eta$ may induce multifractal transitions that cause flux scaling deviates from the naive/dimensional scaling $\ell^{3-\gamma}$ (for instance, Schertzer and Lovejoy (2011) and references therein).

**L335-360**

Regarding the above discussion on the conservation of the mean energy flux, the "computational nightmare" currently envisaged by the authors seems to be:
- either overoptimistic if the mean energy flux is effectively conserved. Indeed, the authors bound below the necessary range of explicit scales by $\eta_{h=h_{min}}$, which only guarantees the presence of singularities that are almost surely present on a unique sample, not the rarest ones that are generated by extreme events;
- or over-pessimistic if the mean energy flux is not conserved: the singularity $h_{min}$ will be not actually reached.

**Conclusions**

...

**References**
Frisch, U., Lesieur, M. and Schertzer, D. (1980) 'Comments on the quasi-normal Markovian approximation for fully-developed turbulence', Journal of Fluid Mechanics, 97(1), pp. 181–192.
Forster, D., Nelson, D. R. and Stephen, M. J. (1977) 'Large-distance and long-time properties of a randomly stirred fluid', Physical Review A, 16, pp. 732–749. doi: 10.1103/PhysRevA.16.732.,
Halsey, T. C. et al. (1986) 'Fractal measures and their singularities: the characterization of strange sets', Physical Review A, 33, pp. 1141–1151.
Herring, J. R. et al. (1982) 'A comparative assessment of spectral closures as applied to passive scalar diffusion', Journal of Fluid Mechanics, 124, pp. 411–437. doi: 10.1017/S0022112082002560.
Hubert, P. et al. (1993) 'Multifractals and extreme rainfall events', Geophysical Research Letters, 20(10), p. 931. doi: 10.1029/93GL01245.
Parisi, G. and Frisch, U. (1985) 'On the singularity structure of fully developed turbulence', in Ghil, M., Benzi, R., and Parisi, G. (eds) Turbulence and predictability in geophysical fluid dynamics and climate dynamics. Amsterdam: North Holland, pp. 84–88.
Richardson, L. F. (1926) 'Atmospheric diffusion shown on a distance-neighbour graph', Proc. Roy. Soc., A110, pp. 709–737.
Schertzer, D. and Lovejoy, S. (1991) 'Nonlinear geodynamical variability: multiple singularities, universality and observables', in Schertzer, D. and Lovejoy, S. (eds) Non-linear variability in geophysics: Scaling and Fractals. Kluwer, pp. 41–82.
Schertzer, D. and Lovejoy, S. (2011) 'Multifractals, Generalized Scale Invariance and Complexity in Geophysics', International Journal of Bifurcation and Chaos, 21(12), pp. 3417–3456.

---

## Author Comment (AC2)

**NPG-2021-19 : Reply to comments from Referee #1**

This study concerns the modeling, through interpretation of laboratory experiments, of the dynamics of the very large features of the Earth system on climate time-scales. The manuscript (presumably) covers the Lecture for the Lewis Fry Richardson (EGU) medal of Berengère Dubrulle, with several of her co-authors. It is entitled "How many modes are needed to predict climate bifurcations? Lessons from an experiment". The paper is placed in the broad framework of the question of whether a scientifically acceptable answer can be given to society concerning the evolution of climate, and what effort (theoretical, modeling, experimental and numerical) is needed in order to be able to do so reliably. Two examples are specifically examined: the switching in the atmosphere between convection and blocking, and the Niño-Niña transition, a prediction of which would be extremely useful to society (and to modelers of course). The principal tool is the experimental facility used by this group in Saclay (France), together with sophisticated mathematical tools. Specifically, the study is focusing on spontaneous transitions of the von Kármán flow between two counter-rotating cylinders under different types of forcing. Reynolds numbers are of the order of 1013 (or less, when using glycerol instead of water), basically too small compared with geophysical flows, but still very enlightening.

**We thank the Referee for her/his/their positive comment.**

In the last sentence of the abstract and in the conclusion, the authors stress the importance of describing very small scales to capture fluctuations of correct intensity and scale. Of course, one thinks of the 50-year old Donnella Meadows et al. project at MIT that predicted a collapse of the world climate and economic system in or around 2020 with a model having a small number of relevant variables and different complex (feedback) interactions. Are these models enough? With such drastic truncations, how are the all-essential small-scales treated or incorporated? One could argue that the equivalence between Earth systems and the von Kármán (VK) flow is not quite as much as stated in the paper; in particular, what of the role of stratification, which is strong at large scale? What is the effect of shear? Can it be quantified? Do they play any role in these transitions? What about chemistry, moisture and the like?

Thank you for this interesting comment pointing at the rich range of scales and phenomena embedded in the climate system. Regarding the first question raised by the reviewer, about the possibility of representing collapse and bifurcation with small number of degrees of freedom, we do believe that the best approach to complexity is to use both simple conceptual models (as the one presented in Faranda et al. 2017, PRL for the VK flow or Faranda et al. 2019 ESD, for the jet stream) as well as complex models (e.g. DNS or LES for the VK and full numerical integration of primitive equations for the climate). While simple models can quickly provide an overview of the bifurcation landscape of a system, fine models can be used to explore targeted regions of the phase space where interesting phenomena produce e.g., extreme dissipation events in the VK flow or convective events in climate simulations. Furthermore, we would be very careful in the new version of the manuscript with suggesting "equivalences" between the VK and the climate system: our idea is rather to show that there is some universality in the behavior of highly turbulent flows in confined geometries. On one hand, the role of shear and stratification mentioned by the reviewer can indeed be observed and quantified in the VK experiment by varying the geometry of the impellers and, as in geophysical flows, those variations can trigger bifurcations. On the other hand, phenomena driven by biogeochemical reactions are beyond the analogy with the von Karman flow. We will make this explicitly clear in the new version of the manuscript in the discussion section.

Also, concerning the computational nightmare, sometimes called the curse of dimensionality, there are several papers claiming that neural networks can lead to avoiding that curse. Do the authors have a comment on that?

Indeed, in recent years there have been developments in understanding that the computational nightmare can be partially solved by applying neural networks or, more generally, machine learning approaches (see, e.g. Pathak et al. for the artificial intelligence applied to the behavior of chaotic systems). We would like to stress that these approaches are never holistic and often target a specific subset of spatial and temporal scales of the climate systems, e.g. the prediction of geophysical data (Wu et al., 2018), the parameterizations of subgrid processes in climate models (Krasnopolsky et al., 2005; Krasnopolsky and Fox-Rabinovitz, 2006; Rasp et al., 2018; Gentine et al., 2018; Brenowitz and Bretherton, 2018, 2019; Yuval and O'Gorman, 2020; Gettelman et al., 2020; Krasnopolsky et al., 2015; Grover et al., 2015; Haupt et al., 2018; Weynet al., 2019) and nowcasting (i.e. extremely short-term forecasting) of weather variables (Xingjian et al., 2015; Shi et al., 2017;Sprenger et al., 2017), the quantification of the uncertainty of deterministic weather prediction (Scher and Messori, 2018). The greatest challenge of entirely replacing the equations of climate models with neural networks capable of producing reliable long and short-term forecasts of meteorological variables is, to the best of our knowledge, not yet achieved with these methods. This has been added to the manuscript.

This paper is very pleasant to read. It describes the thinking going-on at the intersection of several fields of physics and the creativity in the development of new data and in interpreting it. I recommend publication.

We thank the Referee for his/her/their positive comment

**Details:**

1. In the very first sentence, it is a bit odd to see the first author (BD) talk of herself in the third person

We understand the Referee's comment. The manuscript was initially written in the first person, but we had to change it following another comment from the Editor. As a compromise, we have named the first paragraph "context" so that it is clear that it is just the story behind the review and we have written it so that it shows how it lead to the present paper co-signed with 4 other persons.

2. In the title, fry and ?: might have to be changed

We have removed the Lewis Fry Richardson Medal Lecture in the title.

3. I have several problems with Figure 1 (l. 55+). What is a Dansgaard-Oeschger event? What is ka? What are M12345? What do the 8 12 14 18 etc refer to? Also, the insets are way too small, I suggest making the figure substantially larger. A reference (to Didier Paillard or other) would be necessary as well.

Figure 1 has been reworked for improved readability, and the unnecessary information on the graph has been removed. A reference to the original source of the data [K. K. Andersen et al., High resolution record of Northern Hemisphere climate extending into the last interglacial period, Nature 431, 147-151 (2004)] has been added to the text.

4. line 92+, perhaps one could cite and discuss a bit the drastic reduction of modes (down to 3) done by Ed Lorenz and which led to many investigations of lack of predictability of atmospheric flows. For example, why is 3 enough (when, as pointed out by the authors, we need a priori 1024 modes)? Why not 5 or 13? And what do larger systems bring compared to the small (and successful) system? When do we know that we have enough small scales? What criteria can be applied?

We thank the Referee for this suggestion. Indeed, the work of Edward Lorenz provides the lower bound for the minimum number of modes to both obtain chaotic behaviour and reproduce several salient features of Rayleigh-Bénard convection dynamics. We have mentioned this point as well as a few References in Section 2 -- around line 100 in the new version of the manuscript.

5. Figure 2: give the unit of sizes (0.925, 1, ...)

The sizes are provided in units of the cylinder radius R, but we have changed them anyway for more clarity.

6. line 164, ITCZ is not defined

We have added a mention of the full name of the Intertropical Convergence Zone (ICTZ) to the manuscript, and removed the acronym since it is only used once.

7. Figure 3: Is there a significance in the fact that the amplitudes in flow (a) are smaller than for (b) and (c)?

The very strong shear layer produced in the experiments is located away from the impellers in the symmetrical state. In contrast, in bifurcated states, the whole flow rotates in the direction of the dominant impeller, whereas the shear layer is moved towards the slave impeller. As a result, both the angular velocity and the (absolute) angular momentum are significantly higher in bifurcated states than in the symmetrical state. We have mentioned this briefly in the caption of Figure 3.

8. Figure 5: Spring and Summer states are undefined here. One could at least refer to Figure 3, or perhaps add a line in Table 1 defining the 3 (or more) states (Spring or autumn, Austral and Boreal Summers, and Transitional) since they are key ingredients of the analysis and the paper refers to them often.

**We have provided a definition of spring/autumn and austral/boreal summer branches in the caption of Figure 5.**

9. Figure 11: Multifractal spectrum of what?

**We plot here the multifractal spectrum exponent of the wavelet velocity increments. We have clarified this point in the caption of Figure 11.**

10. line 345: Are there not strong velocity gradients in the Planetary Boundary Layer as well as the extreme events mentioned here? Do they not translate into large-scale extremes?

**We agree with the referee. We have rephrased the sentence to make that clearer.**

11. line 351 and beyond:

**The comment from the Referee seems to be missing here.**

12. line 365: By few modes, do you mean 3 as in the Lorenz model, or 104 as in the glycerol experiment described in the paper? Again, how little can it be? How does one determine this threshold? Is not the computational nightmare mentioned in Section 5.4, \*also\* an experimental nightmare?

We mean 104. This has been clarified. Regarding the sequel, we have answered to it previously, see above (comment 4).

---

## Author Comment (AC3)

**NPG-2021-19 : Reply to comments from Referee #2**

*A review of "Lewis Fry Richardson Medal Lecture - How many modes are needed to predict climate bifurcations? : Lessons from an experiment" by Dubrulle et al., NPG-2021-19.*

*General comment This manuscript focuses on a controversial and challenging question: what can we learn about climate dynamics from a laboratory fluid experiment?, and it claims that there is a lot to learn! This optimistic answer is argued using a given generality of dynamical systems, more precisely stochastic attractors, and turbulence phenomenology, namely cascades and finally multifractal intermittency. Moreover, this approach is presented in a nice and elegant way. It certainly helps readers to emerge smoothly within the text, despite some sharp theoretical corners that remain beyond the present paper considerations.*

**We thank the Referee for their positive comment.**

*The paper is presented as a review, which is basically true as far as the work of the team and its laboratories is concerned. However, to my opinion, it would gain from taking into account the earlier findings of other teams although they did not start from a laboratory experiment, but, for example, from the stochastic interactions between weather and climate, both analysed in a multifractal way.*
*A certain number of clarifications seem necessary, in particular on the pivotal issue of the energy flux conservation. A subsequent question is whether all the conclusions would hold in the absence of energy flux conservation.*
*Below is a series of detailed comments and some suggestions which will hopefully be useful for a revision of this challenging manuscript.*

- *Title    It is surprising to see « Lewis Fry Richardson Medal Lecture » in the title, at least because the lecture was given by the Medalist, whereas the paper is cosigned by 4 other persons (see suggestions for L16-28 below).*

  ***We have removed the Lewis Fry Richardson Medal Lecture in the title.***

- *Affiliations    Affiliations 1 and 2 are identical.*

  ***Fixed. Thank you.***

- *L16-28    This paragraph should be rewritten slightly to make it clear that this article is based on the 2021 Lewis Fry Richardson Medal Lecture presented by B. Dubrulle (avoiding to put it in the title) and that both are the result of personal intuitions and collective work.*

**We have named the first paragraph "context" so that it is clear that it is just the story behind the review and we have written it so that it shows how it leads to the present paper co-signed with 4 other people.**

- *L75-78    The current "astronomical number" of degrees of freedom is obtained in a 3D isotropic framework, while even larger number of degrees of freedom (N≈1027) has been obtained in an anisotropic framework (e.g., Schertzer and Lovejoy, 1991), a priori much more relevant for the atmosphere.*

  **We have mentioned anisotropy in the text, changed accordingly the number of degrees of freedom and cited the book by Schertzer and Lovejoy.**

- *L89-93   It should be mentioned that the eddy viscosity has been known since at least Richardson (1926) to be highly scale-dependent, so the quoted estimates are only relevant at given scales that need be specified. Moreover, the action of small scales on larger ones is not limited to eddy viscosity with the presence of a beating / backscatter / renormalized forcing term (e.g., Forster et al, 1977; Frisch et al., 1980; Herring et al. 1982, and references therein). The introduction of an eddy viscosity is far from being a "procedure [which] may sound foolish and bound to failure", whereas it corresponds to a not so sophisticated renormalisation of the Green function that is indispensable to prevent accumulation of energy at the smallest explicit scales and the resulting explosion of the model (see references above). However, it is far from being sophisticated enough, particularly with regard to the necessity to also renormalise forcing and intermittency. So there is no reason to claim that the resulting simulations are that close to reality. In particular, without providing the objective metrics that have been developed to measure it.*

**We have expanded our discussion of turbulent viscosity at the end of Section 2.**

- *Minor note       the XXX surrounding the reference Flato et al., 2013 must be suppressed.*

**This has been fixed in the current version of the manuscript.**

- *Fig2 caption and other places    It would be important to clarify that what is called "angular momentum" is in fact its projection on the vertical axis oriented from bottom to top.*

**We have indicated in the caption of Figure 2 and in Table 1, and in the caption of Figure 4 that the angular momentum we consider is indeed along the vertical axis**

- *L145-152        It might be useful to give more specific names to O(2) and SO(2), e.g. "(general) orthogonal group" and "special orthogonal group" respectively, instead of just the generic name "symmetry group" for both, as well as to recall that SO(2) is the (connected) component of O(2) whose elements have +1 determinants, the other component having determinants -1. This can be particularly useful in understanding the reduction of O(2) into SO(2) with the breakdown of rotational symmetry .*

**These precisions have been added in Section 4.1 of the current manuscript.**

- *L242      It would be useful to make the mentioned stochastic Duffing attractor much less mysterious, for example, that it results from the coupling of the periodic forcing of the classical deterministic Duffing attractor (2D) with a Langevin equation (1D).*

**We have added such a precision in Section 4.5.**

- *L323-325        The figure reference should point to Fig.11b, not Fig.12b.*

   **The local energy transfers are shown in figure 12-b, while figure 11-b shows the multifractal spectrum. We therefore think that the reference is correct as it is.**

- *L323-325        C(h) is rather the (statistical) co-dimension of the support of the singularity h than a multifractal spectrum (Halsey et al., 1986), which originally denoted the (geometrical) dimension f(h)= 3-C(h) introduced by Parisi and Frisch (1985), where C(h) is then constrained to be ≤ 3 (in a 3D embedding space). On the relations between dimension and co-dimension formalisms see, for instance, Schertzer and Lovejoy (2011) and references therein. In particular, singularities h such that C(h) > 3 are almost surely not observable*

*on a unique sample (Hubert et al. 1973). The fact that the minimum of C(h) occurs at $h_0 = 0.35 \approx 1/3$ is rather related to a weak intermittency of the mean energy flux, more precisely to a low co-dimension $C_1$ of the latter: $C_1 = 3h_0 - 1 = 0.05$ with both the assumptions of log-normality and conservation of the energy flux, i.e., the strict scale invariance of the mean energy flux density $\varepsilon_\ell \approx \delta_\ell u^3/\ell \approx \ell^{-\gamma}$; $\gamma = 1 - 3h$. However, this estimate strongly disagrees with the other estimate $C_1 = 1$ obtained from C(0) = 1, which belongs to the empirical parabolic fit C(h) plotted in Fig.11b. This is because this adjustment supports the first assumption, but not the second. Indeed, when obtained from the co-dimension $c_\varepsilon(\gamma)$ of a conservative flux $c_\varepsilon(\gamma) = (\gamma + C_1)^2 / 4C_1$, the function C(h) depends on the unique parameter $C_1$, whereas here it requires two independent parameters (horizontal shift and rescaling) to obtain a vertically orientated parabola. Although there is no obvious theoretical reason compelling that the points (0.35, 0) and (0,1) belong to the curve C(h), this seems to be supported by the empirical extrapolation. Furthermore, the empirical fit is precisely in agreement with the conservation of the statistical moment of order $q_\ell \approx 0.18$ instead of q = 1 and all statistical moments of order diverge with increasing resolution, including the mean flux (q = 1). The latter corresponds to the estimate $h \approx 0.17$ given by the authors as the smallest measurable singularity. I think the current failure to conserve the energy flux needs to be stated clearly and whether it is a system or a model failure needs to be clarified. A similar clarification should be brought on the impossibility to measure singularities h lower/$\gamma$ higher than those of the mean flux, as well as on the nature of this impossibility.*

**The referee would of course be correct if the quantities we had been using for the determination of the scaling exponents would be the flux. In the work of Faller, however, we did not have enough statistics to ensure convergence of the unsigned moments. Hence, we used absolute moments instead. In such a case, we did not have to satisfy the energy flux conservation and so the condition does not hold. This procedure is in fact rather common in  turbulence, and to correct it people sometimes divide their exponent by $\zeta(3)$ to renormalize the result, but we chose not to do that here. Indeed, our even exponents are  correct in any case. We have added this precision in the caption of Figure 11 b and at the beginning of Section 5.2.**

- *L328-333        It is important to distinguish the energy flux $\Pi_\ell$ from its density $\varepsilon_\ell$ (i.e. it is not just a terminology issue): $\Pi_\ell$ results from the 3D integration on a volume $\ell^3$ of its density $\varepsilon_\eta$ at much smaller scales $\eta << \ell$. Strong fluctuations of $\varepsilon_\eta$ may induce multifractal transitions that cause flux scaling deviates from the naive/dimensional scaling $\ell^{3-\gamma}$ (for instance, Schertzer and Lovejoy (2011) and references therein).*

  **We have added a precision regarded this in the corresponding paragraph.**

- *L335-360 Regarding the above discussion on the conservation of the mean energy flux, the "computational nightmare" currently envisaged by the authors seems to be:*
  - *either overoptimistic if the mean energy flux is effectively conserved. Indeed, the authors bound below the necessary range of explicit scales by $\eta_{h=h,min}$ which only guarantees the presence of singularities that are almost surely present on a unique sample, not the rarest ones that are generated by extreme events;*
  - *or over-pessimistic if the mean energy flux is not conserved: the singularity $h_{min}$ will be not actually reached.*

  **Theoretically, the referee is correct, but we think that the present  discussion, taking into account the data at our disposal, offers a reasonable balance in between the two options.**

---

## Referee Report (RR1)

**Comment on the authors' response to my referee's comments**
**NPG-2021-19**

I very much appreciate that the authors ensure that most of my comments are addressed in some way in the current version of the manuscript. However 3 questions remain open:

A.    *Regarding the authors' response to my comment on the figure reference in the following text (initially L325): « We have computed this multifractal spectrum in the von Kármán flow, using both experimental measurements and numerical simulations. The result is shown in Figure 12b. »:*
It is difficult to understand why the authors continue to refer here to local energy transfers (Figure 12b). Therefore, either the figure reference should point to the multifractal spectrum (Figure 11b), as I previously suggested, or the text should be clarified.

B.    *Regarding the authors' response to my comments on the conservation of the energy flux (initially L323-325)*
I do not think that the arguments put forward by the authors can fully satisfy the readers of NPG, at least for the following reasons:
i)   It should be explained why statistics are not sufficient to ensure the convergence of signed moments (assuming "unsigned" was a typo), while they are considered sufficient for absolute moments over the same range of statistical orders.
ii)  I suppose that when mentioning the signed moments, the authors have in mind Kolmogorov's 4/5-law which gives a signed proxy for the energy flux density and whose derivation rigor has sometimes been debated.
iii) Then in particular in the inertial range, the relevance of the unsigned proxy obtained by the absolute moments of the third order could not be completely excluded.
iv)  However, the latter is statistically scale independent for the moment of order $q \approx 0.18$ (instead of $q = 1$) and its moments diverge for higher orders $q$ with increasing resolution, including its mean ($q = 1$).
v)   This high ratio ($\approx 5$) between these two statistical orders may bring into question whether the average signed proxy will be scale independent. For instance, it seems difficult to "correct" this ratio by the renormalisation of the exponents mentioned by the authors, and whose practice (references will be welcome) tends to accredit that there is no empirical evidence of such a large statistical deviation.

Overall, I remain convinced that these facts should be addressed in the revised version. Indeed, a review paper is an opportunity to point out difficulties that could have been overlooked previously. To help the readers of NPG form their own opinions on this challenging question, the already suggested clarifications on the co-dimension formalism should also be taken into account, at least partially. Notably because it was intentionally designed for unsigned quantities.

C.    *Regarding the authors' response to my comments on the "computational nightmare" (initially L335-360):*
It is not so obvious that « a reasonable balance in between the two options » has been struck by the authors, as my comment is not only "theoretically" correct, but it also does not depend on the data availability. I think that the authors could easily start to state their point of view and then mention the possible shortcomings that I mentioned.

---

## Editor Decision (ED1)

Dear Author,

Your paper takes now into account most questions and comments from the Referee 2 reports and introduces some new insights on multifractal intermittency and the multifractal analysis performed. In particular, Fig.12 was improved and it is also helpful to have now an explicit expression for the 2-parameter parabolic approximation of the codimension function $C(h)$ (line 393). This helps to better understand Referee 2's theoretical argument that, to ensure the conservation of the flux of energy, one should use only a 1-parameter approximation.

I am therefore glad to inform you that your paper is accepted up to "technical corrections". I issue only a few suggestions below and recommend a careful reading, In particular the formulation of some inserts could be improved.

Some suggestions
It would be useful to point out, as illustrated by Fig.12b, that the scale parameter $b$ is much more difficult to estimate ($b \approx 0.03$) than the location parameter $h_{mp}$ ($\approx 0.35$), the singularity of maximum probability ($C(h_{mp}) = 0$). Moreover, satisfying that the third-order structure function scales like the scale ($\zeta(3) = 1$, see your online equation at line 395) requires a quite different $b$ value: $b = 1/90 \approx 0.011$. This discrepancy is also illustrated by the fact that the (non-zero) order $q_1$ satisfying the scale independence of the energy flux density statistical moment ($\tau(q_1) = 0$ in eq.4) has the value $q_1 \approx 0.37$ instead of 1. One may note that the estimate $q_1 \approx 0.18$ earlier given by Referee 2 seems to have been obtained with a higher estimate of $b$, while the new Fig.12b can provide more precise estimate.
The aforementioned online equation is in agreement with Referee 2's argument on a unique independent parameter to ensure the conservation of the energy flux, but such a fit seems to be poor and therefore brings into question the parabolic approximation. Therefore, I would suggest more clearly stating that future work is needed to fully clarify this question, rather than the current double negative "the intermittency we measure is not in contradiction…" (line 397). Note also that the sentence that follows is a bit confusing because it suddenly addresses the flux itself, no longer its average.
Please, check the sign of the Kolmogorov 4/3 law (line 349).

Best regards,

Daniel Schertzer (editor)

---

## Author Response (AR2)

**NPG-2021-19 : Reply to comments from Referee #2**

*I very much appreciate that the authors ensure that most of my comments are addressed in some way in the current version of the manuscript.*

**We thank the Referee for their positive comment.**

*However 3 questions remain open:*

> *Point A: L323-325*     *It is difficult to understand why the authors continue to refer here to local energy transfers (Figure 12b). Therefore, either the figure reference should point to the multifractal spectrum (Figure 11b), as I previously suggested, or the text should be clarified..*

> **The referee is right. We are sorry about our previous response, we thought he referred to another figure (the true figure 12b). We have now made the correction.**

*Point B: Regardind the author's respons to my comments on energy flux: I do not think that the arguments put forward by the authors can fully satisfy the readers of NPG, at least for the following reasons:*

i)     *It should be explained why statistics are not sufficient to ensure the convergence of signed moments (assuming "unsigned" was a typo), while they are considered sufficient for absolute moments over the same range of statistical orders.*

ii)    *I suppose that when mentioning the signed moments, the authors have in mind Kolmogorov's 4/5-law which gives a signed proxy for the energy flux density and whose derivation rigor has sometimes been debated.*

iii)   *Then in particular in the inertial range, the relevance of the unsigned proxy obtained by the absolute moments of the third order could not be completely excluded.*

iv)   *However, the latter is statistically scale independent for the moment of order $q\_ \_ \approx \_0.18$ (instead of $q\_ \_= 1$) and its moments diverge for higher orders $q\_ \_$with increasing resolution, including its mean ($q\_ \_= 1$).*

v)    *This high ratio ($\approx 5$) between these two statistical orders may bring into question whether the average signed proxy will be scale independent. For instance, it seems difficult to "correct" this ratio by the renormalisation of the exponents mentioned by the authors, and whose practice (references will be welcome) tends to accredit that there is no empirical evidence of such a large statistical deviation. Overall, I remain convinced that these facts should be addressed in the revised version. Indeed, a review paper is an opportunity to point out difficulties that could have been overlooked previously. To help the readers of NPG form their own opinions on this challenging question, the already suggested clarifications on the co-dimension formalism should also be taken into account, at least partially. Notably because it was intentionally designed for unsigned quantities.*

**We have enlarged the discussion, to explain the difference between signed and unsigned velocity increments, developed more the link with Kolmogorov 4/3 law and stressed that the codimension interpretation was made for unsigned quantities. We have also added new data to the Figure 11b, and added a discussion regarding h_min**

*Point C: Regarding the author's response to my comment on the computational nightmare*

*It is not so obvious that « a reasonable balance in between the two options » has been struck by the authors, as my comment is not only "theoretically" correct, but it also does not depend on the data availability. I think that the authors could easily start to state their point of view and then mention the possible shortcomings that I mentioned.*

**We have done what is suggested by referee and described the two optimistic and pessimistic options.**